# Should altitudinal gradients of temperature and precipitation inputs be inferred from key parameters in snow-hydrological models?

## Denis Ruelland

CNRS, HydroSciences Montpellier, University of Montpellier, Place E. Bataillon, 34395 Montpellier Cedex 5, France

*Correspondence to: denis.ruelland@umontpellier.fr*

**Abstract.** This paper evaluates whether snow-covered area and streamflow measurements can help assess altitudinal gradients of temperature and precipitation in data-scarce mountainous areas more efficiently than using the usual interpolation procedures. A dataset covering 20 Alpine catchments is used to investigate this issue. Elevation dependency in the meteorological fields is accounted for using two approaches: (i) by estimating the local and time-varying altitudinal
gradients from the available gauge network based on deterministic and geostatistical interpolation methods with an external drift; and (ii) by calibrating the local gradients using an inverse snow-hydrological modelling framework. For the second approach, a simple 2-parameter model is proposed to target the temperature/precipitation-elevation relationship and to regionalise air temperature and precipitation from the sparse meteorological network. The coherence of the two approaches is evaluated by benchmarking several hydrological variables (snow-covered area, streamflow) computed with snow-
hydrological models fed with the interpolated datasets and checked against available measurements. Results show that accounting for elevation dependency from scattered observations when interpolating air temperature and precipitation cannot provide sufficiently accurate inputs for models. The lack of high-elevation stations seriously limits correct estimation of lapse rates of temperature and precipitation, which, in turn, affects the performance of the snow-hydrological simulations due to imprecise estimates of temperature and precipitation volumes. Instead, retrieving the local altitudinal gradients using
an inverse approach enables increased accuracy in the simulation of snow cover and discharge dynamics, while limiting problems of over-calibration and equifinality.

## 1. INTRODUCTION

### 1.1. Providing accurate meteorological inputs in mountainous regions

Regionalising air temperature and precipitation is a critical step in producing accurate areal inputs for hydrological models in
high altitude catchments. The ability to correctly reproduce areal precipitation is vital to avoid the failure of hydrological models, which are sensitive to input volumes at the catchment scale (e.g. Oudin et al., 2006; Nicótina et al., 2008). Accurate temperature fields are also particularly important in mountainous regions because temperature is the main driver for snow/rain partition and snowmelt, and consequently influences seasonal discharge (e.g. Hublart et al., 2015; 2016).

However, in areas with complex topography, the characteristic spatial scales of temperature and precipitation estimates
are typically poorly captured, notably when the network of measurements used is sparsed. Gridded datasets obtained by interpolating measurements taken at meteorological stations are thus affected by inaccuracies, which are spatially and temporally variable and difficult to quantify (Haylock et al., 2008; Isotta et al., 2015). Measurement errors depend on local conditions and increase with terrain elevation, as the operational conditions become more extreme (Frei and Schär, 1998). In the case of precipitation, a well-known problem arises from the systematic errors associated with precipitation under-catch
during snowfall (Strasser et al., 2008), especially in windy conditions (Sevruk, 2005). In addition, temperature and precipitation are under-sampled at high elevations, because meteorological stations are mainly located at low elevations for logistical reasons (Hofstra et al., 2010). This makes it difficult to derive the local and seasonal relationship between

meteorological observations and topography, even though this is indispensable for accurate spatial temperature and precipitation estimates (Masson and Frei, 2014). Indeed, atmospheric uplift caused by relief tends to increase precipitation

with elevation through the so-called orographic effect (Barry and Chorley, 1987). Nevertheless, precipitation accumulation trends can show considerable scatter with altitude depending on the region's exposure to wind and synoptic situations (Sevruk, 1997). The relationship between temperature and elevation is generally more obvious. The rate at which air cools with a change in elevation ranges from about -0.98 °C $(100m)^{-1}$ for dry air (i.e., the dry-air adiabatic lapse rate) to about -0.40 °C $(100m)^{-1}$ (i.e., the saturated adiabatic lapse rate; Dodson and Marks, 1997). Average temperature gradients of -

0.60 °C (Dodson and Marks, 1997) or -0.65 °C $(100m)^{-1}$ (Barry and Chorley, 1987) are often used when high precision is not required. However, such average values are known to be rough approximations which are not suitable for more precise studies (see e.g. Douguédroit and De Saintignon, 1984). Notably they mask significant variations in different meteorological conditions and in different seasons. For instance, temperature lapse rates are generally lower in winter than in summer, as shown by Rolland (2003) for Alpine regions.

## 1.2. Schemes for spatial interpolation of air temperature or precipitation

The mapping of air temperature and precipitation using discrete observations based on gauge networks has been extensively studied. Readers can refer to, for instance, Ly et al. (2013) for a review on the different deterministic and geostatistical methods designed for operational hydrology and hydrological modelling at the catchment scale.

Schemes for spatial interpolation of meteorological variables vary in three ways (Stahl et al., 2006): (1) the model used

to characterise the spatial variation of the variable of interest, (2) the method used to choose the surrounding points (number or distance, angular position relative to the prediction point) and (3) the approach used to adjust for elevation. The simplest approach is to choose the nearest station and adjust for elevation according to an assumed lapse rate. However, this method is fairly crude and ignores fine-scale spatial variations. Where more than one station is used in the prediction, a model is required to determine how to interpolate from them. Interpolation weights have been estimated using approaches including

inverse distance weighting (IDW) (e.g. Dodson and Marks, 1997; Shen et al., 2001; Frei, 2014) and geostatistical methods based on kriging (e.g. Garen and Marks, 2005; Spadavecchia and Williams, 2009). Kriging relies on statistical models involving autocorrelation, which refers to the statistical relationships between measured points. Ordinary kriging (OKR) is well-known among kriging algorithms (see e.g. Goovaerts, 1997 for a detailed presentation of these algorithms). Different methods have been developed to deal with the statistical relationship between temperature/precipitation and elevation like

regression analysis (Drogue et al., 2002), or more elaborate geostatistical techniques including simple kriging with local means, kriging with external drift (KED) and co-kriging (CKR): see Goovaerts (2000) for a comparison of these approaches. Among these techniques, KED has been widely used to generate temperature and precipitation maps. For instance, Masson and Frei (2014) showed that KED led to much smaller interpolation errors than linear regressions in the Alps. This was achieved with a single predictor (local topographic height), whereas the incorporation of more extended predictor sets (slope,

circulation-type dependence of the relationship, inclusion of a wind-aligned predictor) enabled only marginal improvement. For daily precipitation, interpolation accuracy improved considerably with KED and the use of a simple digital elevation compared to OKR (i.e., with no predictor). These results confirm that accounting for topography is important for spatial interpolation of daily precipitation in high-mountain regions. Conversely, other authors showed that, even though taking topography into account was indispensable for temperature reconstruction whatever the temporal resolution, it was less clear

for daily precipitation. For example, Ly et al. (2011) reported no improvement in precipitation estimated at a daily time scale if topographical information was taken into account with KED and CKR, compared to simpler methods such as ORK and IDW. In a recent and very complete comparative study, Berndt and Haberlandt (2018) analysed the influence of temporal resolution and network density on the spatial interpolation of climate variables. They showed that KED using elevation

performed significantly better than ORK for temperature data at all temporal resolutions and station densities. For precipitation, using elevation as additional information in KED improved the interpolation performance at the annual time scale, but not at the daily time scale.

Theoretically, KED can account for local differences in topographic influence in different seasons and synoptic situations. Indeed, the regression coefficients computed between the primary variable (temperature or precipitation) and the secondary variable (elevation) are implicitly estimated through the kriging system within each search neighbourhood (Goovaerts, 2000). The relation between variables is thus assessed locally, meaning changes in correlation across the study area can be taken into account. However, as suggested by Stahl et al. (2006) concerning temperature and by Ly et al. (2011) concerning precipitation, care should be taken in applying KED when interpolating daily variables with very few neighbouring sample points. Indeed, methods that compute local lapse rates from the surrounding control points can perform poorly in regions with insufficient high-elevation data, due to inaccurate estimation of local lapse rates.

## 1.3. Placing meteorological fields in a hydrological perspective

A subject that requires further investigation is which methods that produce daily temperature and precipitation fields can provide the best snow cover and streamflow simulations. The usual cross validation for the inter-comparison of interpolation methods is limited, especially in ungauged areas, like the highest parts of mountainous areas. As stressed by Gottardi et al. (2012), a method can perform well in interpolation (at the ground network altitudes) but poorly in extrapolation (higher). This is because the observed set is not representative of the entire feature space. As a result, estimations at high elevations are difficult to check due to the lack of meteorological data. To go further, the use of other data like streamflow measurements may be a good alternative way to validate temperature and precipitation estimations at high-elevation sites.

To date, few studies have compared the performance of different interpolation methods evaluated by hydrological modelling in mountainous areas. Among the few that have, Tobin et al. (2011) showed that kriging (and more specifically KED) can be used effectively to estimate temperature and precipitation fields in complex alpine topography during flood events. Their comparative analyses of the different interpolation techniques suggested that geostatistical methods performed better than IDW. In particular, with elevation as auxiliary information, KED gave the overall best validation statistics for the set of events under study. However, it can be hypothesised that, in many mountainous areas, gauge observations do not include sufficient information to accurately account for the elevation dependency of air temperature and precipitation using interpolation techniques, which are thus limited to providing accurate inputs for snow-hydrological models. On the other hand, numerous calibration parameters controlling snow accumulation (the temperature threshold between the solid and liquid phase, temperature range of phase separation, snowfall gauge under-catch factor) and melt (temperature threshold for snowmelt, degree-day melt factor, snowpack thermal state, etc.) have been introduced in most of the snow accounting routines (SAR) used in operational hydrology: see e.g. HBV (Bergström, 1975), MOHYSE (Fortin and Turcotte, 2007), CEMANEIGE (Valéry et al., 2014), MORDOR (Garavaglia et al., 2017). The aim of using these parameters is to adapt to local snow processes, but they could be used primarily to compensate for errors in the input data without satisfactorily achieving it.

## 1.4. Inverting the hydrological cycle

In contrast, inverting the hydrological cycle with snow-hydrological models may help identify the dependency of the areal inputs on elevation more realistically and enable more accurate snow-hydrological simulations, while simultaneously limiting the number of free parameters. The idea is not completely new and was notably introduced by Valéry et al. (2009) in an attempt to use streamflow measurements to improve knowledge of yearly precipitation in data-sparse mountainous regions. Their results suggested that it was possible to unambiguously identify the altitudinal precipitation gradients from streamflow at a yearly time scale. In another paper, Valéry et al. (2010) proposed regionalisation of daily air temperature and

precipitation to better estimate inputs over high-altitude catchments as regards to the water balance. In their conclusion, the authors claimed that their regionalisation approach also significantly improved the performance of a rainfall-runoff model at a daily time scale. However, the lapse rates in the temperature and precipitation inputs were estimated from gauge observations at the regional scale based on a leave-one-out procedure. This leaves room for potential improvement by locally inferring the lapse rates based on inverse modelling applied at the catchment scale.

A few studies proposed approaches to estimate lapse rates based on hydrological modelling in specific catchments. Zhang et al. (2013) showed that the runoff simulation results involving snowmelt and rainfall runoff were highly sensitive to the temperature and precipitation lapse rates in a Tibetan catchment. Rahman et al. (2014) calibrated the SWAT model in a snow-dominated basin in the Swiss Alps and found also that temperature lapse rate was significantly important for hydrological performance. Naseer et al. (2019) considered a dynamic lapse rate based on a vertical profile of temperature in a catchment in Japan and succeeded to improve the precipitation phase in a distributed hydrological modelling framework. Henn et al. (2016) investigated the value of snow data to constrain the inference of precipitation from streamflow, using lumped hydrologic models and an elevation-band snow model in a Californian basin. Their results suggested that multiple types of hydrologic observations, such as streamflow and SWE, may help to constrain the water balance of high-elevation basins. Le Moine et al. (2015) proposed a calibration strategy where the parameters of both an interpolation model and a daily snow-hydrological model are jointly inferred in a multi-variable approach applied in a catchment in the French Alps. Using a hydro-meteorological modelling chain involving 31 calibrated parameters, they showed the potential of using different types of observations (rain gauges, snow water equivalent measurements and streamflow data) to help assess temperature and precipitation lapse rates according to different weather types. These studies encourage testing whether an inverse modelling approach based on calibrated constant lapse rates can perform well in numerous basins with parsimonious conceptual models.

Improvement is also to be expected from the use of observations such as remotely-sensed snow cover data to calibrate and validate models in addition to the runoff measurements, which can help better assess the reliability of the modelled snow processes (see e.g. Parajka and Blöschl, 2008; Thirel et al., 2013). Moreover, other authors (Franz and Karsten, 2013; He et al., 2014; Riboust et al., 2019) showed that adding snow data information to the calibration procedure enabled the identification of more robust snow parameter sets by making the snow models less dependent on the rainfall-runoff model with which they are coupled. Using both streamflow and snow-cover observations in an inverse modelling approach could thus provide further insights into the most relevant snow parameters, while improving our knowledge of the altitudinal temperature and precipitation gradients in data-sparse mountainous regions.

## 1.5. Objectives

Based on the above issues, this paper investigates whether altitudinal gradients should be inferred from available gauge information when interpolating air temperature and precipitation, or from key-parameters of snow-hydrological models in mountainous areas. To address this question, we use a large dataset of mountainous, snow-affected catchments in the French Alps and we propose a framework to assess the hydrological coherence of gridded datasets and for inferring orographic gradients based on snow-hydrological observations. The rest of the paper is organised as follows. Section 2 describes the study region, the data and their pre-processing. Section 3 provides a brief description of the interpolation procedures tested. Section 4 presents the model assessment methodology. The results are presented and discussed in Section 5, and the main findings, recommendations and future outlooks are summarised in Section 6.

## 2. STUDY AREA AND DATASET

### 2.1. Meteorological data

The study was carried out in the French Alps whose altitudes range from 79 to nearly 4800 m a.s.l. (Fig. 1). A dataset of 78 temperature gauges and 148 precipitation gauges was gathered from the RADOME (*Réseau Automatisé d'Observations Météorologiques Etendues*) database of Météo-France (https://publitheque.meteo.fr) for six administrative departments (*Alpes-de-Haute-Provence*, *Hautes-Alpes*, *Alpes-Maritimes*, *Isère*, *Savoie* and *Haute-Savoie*). The extracted series are the mean daily air temperature and the daily liquid equivalent water depth of total precipitation for each station over an 18-year

period from the 1st of September 1998 to the 31st of August 2016. These gauges were selected because no gap was originally present in their time series from the 1st September 2000 to the 31st of August 2016, thus allowing a coherent and stable signal to be represented over the 16-year period of analysis. The corresponding gauge density is ~3 stations per 1000 km² for temperature and ~5 stations per 1000 km² for precipitation, which is close to the recommended minimum density for mountainous areas (~4 stations per 1000 km², WMO, 2008). Although the spatial distribution of the available meteorological

stations is reasonably balanced, high altitudes (above 2000 m a.s.l.), which represent approximately 20% of the whole study area and 45% of the catchment surface area (Fig. 1b), are not monitored, as temperature and precipitation gauges are mainly located at low and mid elevations: between 235 and 2105 m for temperature, and between 235 and 2006 m for precipitation.

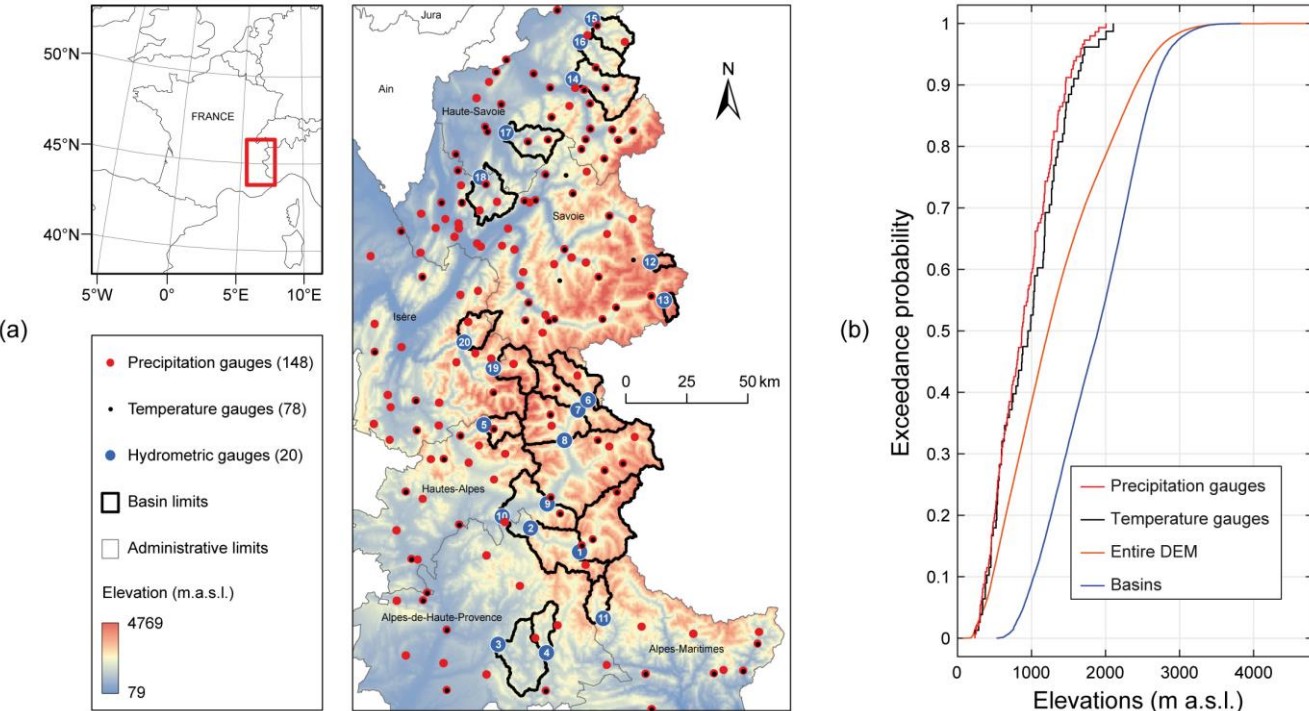

**Fig. 1** Study area and data: (a) Location of the selected precipitation, temperature and streamflow stations, as well as elevations from a SRTM digital elevation model (DEM resampled to a grid with 0.5x 0.5 km cells) in the French Alps; (b)  Elevation distributions of in-situ stations, DEM and basins. The station numbers refer to Table 1.

### 2.2. Streamflow data

In order to avoid case-specific results, a dataset of 20 catchments was gathered from the French hydrological database

(www.hydro.eaufrance.fr) over the study area (Fig. 1 and Table 1). The catchments were selected based on the following criteria: (i) their streamflow regime is considered to be natural since they are located upstream from any major hydraulic installations, such as dams and water transfers; (ii) their streamflow regime is moderately to strongly affected by snow; and (iii) their streamflow series present good quality measurements according to the hydrological reports, with less than 10%

daily missing values for the period 2000–2016. The catchments are located on high reliefs (the median range of altitude is around 2000 m a.s.l.) and are differently affected by snow (between 15% and 66% of their total precipitation falls in the solid form). During the catchment selection process, we tried avoiding glacierized catchments to minimize possible interactions with non-snow related processes that could also influence streamflow. Therefore, most catchments have zero or limited glacierized areas. Catchment areal precipitation and temperature were estimated after inferring altitudinal gradients, as detailed in the current paper. The dataset cover a large range of hydrological conditions with mean annual precipitation, temperature and streamflow ranging from 811 to 2315 mm, -2.3 °C to +8.9 °C, and 344 mm to 1771 mm, respectively.

**Table 1** Streamflow gauging stations and main catchment characteristics. Percentages of glacierized area were estimated from the World Glacier Inventory (NSIDC, 2012). Mean annual precipitation ($P$), snowfall fraction ($S$) and temperature ($T$) were estimated after calibrating local altitudinal gradients over 2000–2016 using the snow-hydrological inverse approach proposed in the current paper (see Test #4 in Table 5).

| # | Station | River | Area | Glacierized area | Elevations (m.a.s.l.) | | Mean annual precip. ($P$) | Snowfall fraction ($S$) | Mean annual temp. ($T$) | Mean annual streamflow ($Q$) |
|---|---------|-------|------|------------------|---------|---------|------|------|------|------|
| | | | (km²) | (%) | Min | Max | (mm/year) | (%) | ( °C) | (mm/year) |
| 1 | Barcelonnette | Ubaye | 549 | 0 | 1132 | 3308 | 856 | 49 | 1.9 | 521 |
| 2 | Lauzet-Ubaye | Ubaye | 946 | 0 | 790 | 3308 | 979 | 45 | 3.0 | 654 |
| 3 | Beynes | Asse | 375 | 0 | 605 | 2273 | 921 | 15 | 8.9 | 344 |
| 4 | Saint-André | Issole | 137 | 0 | 931 | 2392 | 1013 | 23 | 7.0 | 481 |
| 5 | Villar-Lourbière | Séveraisse | 133 | 4 | 1023 | 3623 | 1781 | 49 | 2.4 | 1317 |
| 6 | Val-des-Prés | Durance | 207 | 0 | 1360 | 3059 | 847 | 55 | 0.9 | 688 |
| 7 | Briançon | Durance | 548 | 1 | 1187 | 3572 | 811 | 52 | 1.6 | 714 |
| 8 | Argentière-la-Bessée | Durance | 984 | 3 | 950 | 4017 | 1064 | 52 | 2.3 | 765 |
| 9 | Embrun | Durance | 2170 | 2 | 787 | 4017 | 1075 | 48 | 3.1 | 693 |
| 10 | Espinasses | Durance | 3580 | 1 | 652 | 4017 | 1054 | 45 | 3.5 | 654 |
| 11 | Villeneuve | Var | 132 | 0 | 926 | 2862 | 1072 | 37 | 4.6 | 650 |
| 12 | Val-d'Isère | Isère | 46 | 9 | 1831 | 3538 | 1245 | 63 | -1.5 | 1119 |
| 13 | Bessans | Avérole | 45 | 12 | 1950 | 3670 | 1635 | 66 | -2.3 | 1311 |
| 14 | Taninges | Giffre | 325 | 0 | 615 | 3044 | 2315 | 35 | 5.0 | 1771 |
| 15 | Vacheresse | Dranse d'Abondance | 175 | 0 | 720 | 2405 | 1671 | 30 | 4.8 | 1088 |
| 16 | La Baume | Dranse de Morzine | 170 | 0 | 690 | 2434 | 1637 | 32 | 4.6 | 1285 |
| 17 | Dingy-Saint-Clair | Fier | 223 | 0 | 514 | 2545 | 1649 | 26 | 6.6 | 1243 |
| 18 | Allèves | Chéran | 249 | 0 | 575 | 2157 | 1486 | 23 | 6.9 | 819 |
| 19 | Mizoën | Romanche | 220 | 9 | 1057 | 3846 | 1369 | 56 | 0.9 | 978 |
| 20 | Allemond | L'Eau Dolle | 172 | 2 | 713 | 3430 | 1553 | 47 | 2.4 | 1164 |

## 2.3. Snow cover data

MOD10A1 (Terra) and MYD10A1 (Aqua) snow products version 5 were downloaded from the National Snow and Ice Data Center for the period 24 February 2000–1 January 2017. This corresponds to 6157 dates among which 98.8% are available for MOD10A1 and 85.8% for MYD10A1 since Aqua was launched in May 2002 and became operational in July 2002. These snow products are derived from a Normalised Difference Snow Index (NDSI) calculated from the near-infrared and green wavelengths, and for which a threshold was defined for the detection of snow (Hall et al., 2006, 2007). Cloud cover represents a significant limit for these products, which are generated from instruments operating in the visible-near-infrared wavelengths. As a result, the grid cells were gap filled to produce daily cloud-free snow cover maps of the study area. The different classes in the original products were first merged into three classes: no-snow (no snow or lake), snow (snow or lake ice), no-data (clouds, missing data, no decision, or saturated detector). The missing values were then filled according to a gap-filling algorithm inspired by techniques proposed in the literature (Parajka and Blöschl, 2008; Gafurov and Bárdossy, 2009; Gascoin et al., 2015). The algorithm works in three sequential steps:

(i)      Aqua/Terra combination: for every pixel, if no-data was found in MOD10A1 then the value from MYD10A1 was used instead. Priority was given to MOD10A1 because MYD10A1 was found to be less accurate (see Gafurov and Bárdossy, 2009).

(ii)     Temporal deduction by sliding time filter: a no-data pixel was reclassified as snow (no-snow) if the same pixel was classified as snow (no-snow) in both the preceding and following grids. The preceding and following grids were searched within a sliding temporal window, whose size was incremented up to 9 days in order to reduce the remaining fraction of no-data pixels to below 10% (Fig. 2a). It should be noted that three periods of gaps in an upper time window (11, 13 and 18 days) were present in the data because of technical failures of the MODIS sensor. In these cases, a longer time deduction was used beforehand to specifically fill these periods.

(iii)    Spatial deduction based on elevation and neighbourhood filter: for each date and each pixel, a 3x3 neighbourhood spatial filter was used to account for the elevation and the data in the neighbouring pixels to fill the remaining no-data pixels. Two configurations were considered: either the central pixel has no-data and the algorithm tries to attribute a neighbouring value, or the central pixel has a value that can be assigned to some of its neighbours. The two configurations were repeated until there were no more gaps (Fig. 2a).

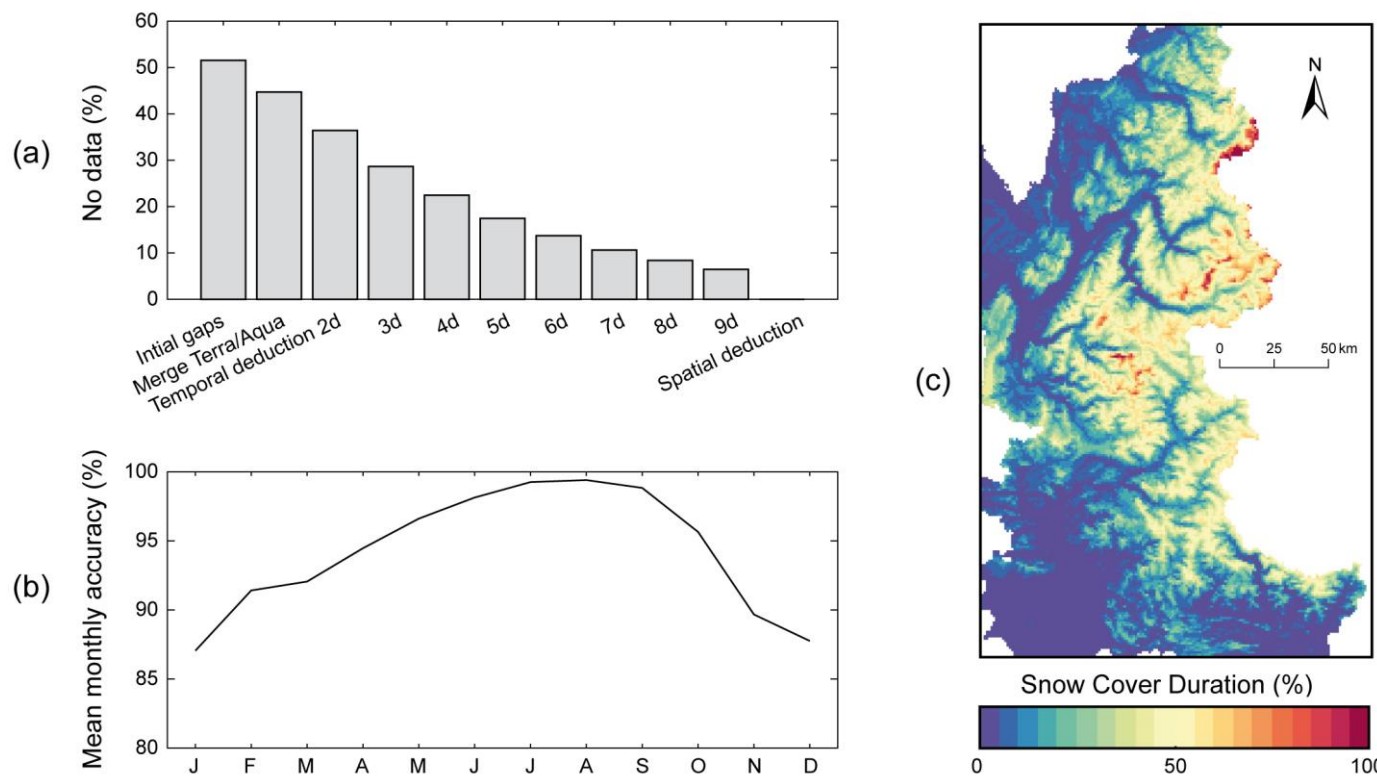

**Fig. 2** Results of gap-filling applied to MODIS snow products: (a) Evolution of the number of pixels classified as no-data (e.g., clouds) during the gap-filling procedure; (b) Mean monthly accuracies according to validation based on confusion matrices with 1 image/month, i.e. ~200 cloud-free images over the 2000–2016 period; (c) Snow cover duration based on gap-filled MODIS snow products over the 2000–2016 period.

The resulting database consists of 5844 binary (snow/no-snow) maps at 500 m spatial resolution for the period 2000–2016 (16 hydrological years, from the 1st of September 2000 to the 31st of August 2016). As a synthesis of these maps, snow cover durations over the study area are presented in Fig. 2c.

In order to validate the gap-filling technique, a daily snow product with less than 10% of no-data pixels was selected for each month of the studied period. These images were "blackened" (i.e. with 100% no-data pixels), before applying the

algorithm over the entire period to fill all gaps, including validation images. Filling accuracy was estimated for each image removed by computing confusion matrices which compared the pixels of the removed validation images and the filler reconstructions of these images. Validation based on confusion matrices with 1 image/month showed that the gap-filling technique applied to the MODIS snow-products led to the reconstruction of images with average accuracies of 94% (Fig. 2b). The mean monthly accuracies show greater ease in filling gaps in summer than in winter due to the differences in cloud obscuration. However it should be noted that the actual accuracy of the MODIS gap-filling technique is necessarily greater than that of the validation procedure, in which many quality images needed to fill the gaps were missing.

## 3. INTERPOLATION PROCEDURES

This section briefly presents the different spatial estimators used in the present study. The interpolation methods analysed include inverse distance weighted (IDW), ordinary kriging (ORK), kriging with external drift (KED) and IDW with external drift (IED). Interest readers can refer to Goovaerts (2000) for a detailed presentation of the different kriging algorithms, and to Diggle and Ribeiro (2007) for their implementation in the public domain in the *GeoPackage* in *R*.

### 3.1. Spatial interpolation methods

3.1.1. Inverse distance weighting

Let us consider the problem of estimating the given variable $z$ at an unsampled location $u$ using only surrounding observation data. Let $\{z(u_\alpha), \alpha = 1, \ldots, n\}$ be the set of data measured at $n$ surrounding locations $u_\alpha$. The inverse distance weighting (IDW) method estimated $z$ as a linear combination of $n(u)$ surrounding observations with the weights being inversely proportional to the square $\omega$ distance between observations and $u$:

$$Z_{\text{IDW}}(u) = \frac{1}{\sum_{\alpha=1}^{n(u)} \lambda_\alpha(u)} \sum_{\alpha=1}^{n(u)} \lambda_\alpha(u) z(u_\alpha) \quad \text{with} \quad \lambda_\alpha(u) = \frac{1}{|u - u_\alpha|^\omega} \tag{1}$$

The basic idea behind the weighting scheme is that observations that are close to each other on the ground tend to be more alike than those located further apart, hence observations closer to $u$ should receive a larger weight.

3.1.2. Ordinary kriging

Instead of Euclidian distance, geostatistics uses the semivariogram as a measure of dissimilarity between observations. The experimental semivariogram is computed as half the average squared difference between the components of data pairs:

$$\hat{\gamma}(h) = \frac{1}{2N(h)} \sum_{\alpha=1}^{N(h)} [z(u_\alpha) - z(u_\alpha + h)]^2 \tag{2}$$

where $N(h)$ is the number of pairs of data locations a vector $h$ apart. The hypotheses of spatial variability were here homogeneity and an isotropic spatial pattern due to the lack of sufficient sampled points, and hence identical variability in all directions.

Kriging is a generalized least-squares regression technique that makes it possible to account for the spatial dependence between observations, as revealed by the semivariogram, in spatial prediction. Like the inverse distance weighting method, ordinary kriging (ORK) estimates the unknown variable $z$ at the unsampled location $u$ as a linear combination of neighbouring observations:

$$Z_{ORK}(u) = \sum_{\alpha=1}^{n(u)} \lambda_\alpha^{ORK}(u) z(u_\alpha) \quad \text{with} \quad \sum_{\alpha=1}^{n(u)} \lambda_\alpha^{ORK}(u) = 1 \tag{3}$$

The ordinary kriging weights $\lambda_\alpha^{ORK}(u)$ are determined such as to minimise the estimation variance $Var\{Z_{ORK}(u) - z(u)\}$, while ensuring the unbiasedness of the estimator $E\{Z_{ORK}(u) - z(u)\} = 0$. These weights are obtained by solving a system of linear equations known as the ordinary kriging system:

$$\begin{cases} \sum_{\beta=1}^{n(u)} \lambda_\beta(u)\gamma(u_\alpha - u_\beta) - \mu(u) = \gamma(u_\alpha - u) & \alpha = 1,\dots,n(u) \\ \sum_{\beta=1}^{n(u)} \lambda_\beta(u) = 1 \end{cases} \tag{4}$$

where $\mu(u)$ are Lagrange parameters accounting for the constraints on the weights. The only information required by the kriging system (4) are semivariogram values for different lags, and these are readily derived once a semivariogram model has been fitted to experimental values. In this study, we dealt with the fitting of the semivariogram using two existing theoretical models, as presented below:

- Exponential model

$$\gamma(h;\theta) \begin{cases} 0, & h = 0, \\ \theta_0 + \theta_1[1 - \exp(-3(\|h\|/\theta_2)], & h \neq 0, \end{cases} \tag{5}$$

for $\theta_0 \geq 0, \ \theta_1 \geq 0 \ and \ \theta_2 \geq 0$.

- Spherical model

$$\gamma(h;\theta) \begin{cases} 0, & h = 0, \\ \theta_0 + \theta_1\left(\dfrac{3\|h\|}{2\theta_2} - \dfrac{1}{2}\left(\dfrac{\|h\|}{\theta_2}\right)^3\right), & 0 < \|h\| \leq \theta_2, \\ \theta_0 + \theta_1, & h > \theta_2, \end{cases} \tag{6}$$

for $\theta_0 \geq 0, \ \theta_1 \geq 0 \ and \ \theta_2 \geq 0$.

The spherical model was tested because it is the most widely used semivariogram model and is characterised by linear behaviour (Goovaerts, 2000). The exponential model was selected in addition because it is recommended in the literature for spatial analysis of temperature (Tobin et al., 2011) and precipitation (Bárdossy and Pegram, 2013; Masson and Frei, 2014) in high-mountain regions. Each of these models was combined with a nugget effect, sill and range as parameters. An automatic procedure was necessary to fit the semivariogram model to experimental values over the study period (1998–2016). The models were fitted using regression such that the weighted sum of squares of differences between the experimental and model semivariogram is minimum (see Goovaerts, 2000).

### 3.2. Accounting for elevation dependency

3.2.1. Kriging with external drift

Kriging with an external drift (KED) predicts sparse variables which are poorly correlated in space by considering that there is a local trend within the neighbourhood; primary data is assumed to have a linear relation with auxiliary information exhaustively sampled over the study area (Ahmed and de Marsily, 1997). KED thus uses secondary information (such as elevation) to derive the local mean of the primary attribute $z$ and then performs kriging on the corresponding residuals:

$$Z_{\text{KED}}(u) - m_{\text{KED}}(u) = \sum_{\alpha=1}^{n(u)} \lambda_{\alpha}^{KED}(u)[z(u_{\alpha}) - m_{\text{KED}}(u_{\alpha})]$$

(7)

with $\quad m_{\text{KED}}(u) = a_0(u) + a_1(u)y(u)$

where $y(u)$ are elevation data available at all estimation points, $a_0$ and $a_1$ are two regression coefficients estimated from the set of collocated variable of interest and elevation data $\{z(u_{\alpha}), y(u_{\alpha}), \alpha = 1, \dots, n\}$ using a simple linear relation.

 The KED procedure was applied at each time step independently and within each search neighbourhood when the time series were interpolated. The coefficients $a_0$ and $a_1$ thus varied in space and time, which makes possible to consider a variable space-time relationship between the primary variable (temperature or precipitation) and the secondary variable (elevation).

3.2.2. Inverse distance weighting with external drift (IED)

The external drift approach was also tested using the inverse distance weighting procedure to propose an original technique, which we called IDW with external drift (IED), as follows:

$$Z_{\text{IED}}(u) - m_{\text{IED}}(u) = \frac{1}{\sum_{\alpha=1}^{n(u)} \lambda_{\alpha}(u)} \sum_{\alpha=1}^{n(u)} \lambda_{\alpha}(u)[z(u_{\alpha}) - m_{\text{IED}}(u_{\alpha})]$$

(8)

with $\quad m_{\text{IED}}(u) = a_0(u) + a_1(u)y(u)$

### 3.3. Leave-one-out procedure

The interpolation parameters ($n(u)$ and $\omega$ for IDW and IED, and $n(u)$ and theoretical models for ORK and KED) were calibrated and the interpolation performance was assessed by "leave-one-out" cross validation, which consists of the following principle: a successive estimation of all sampled locations was performed by using all other stations while always excluding the sample value at the location concerned. The spatial models were validated against RMSE (root mean square error) for temperature and precipitation at daily, monthly and yearly time scales. Since the external drift computation and kriging weights can sometimes lead to negative precipitation amounts (Deutsch, 2006), *a posteriori* correction was performed to replace all negative-estimated precipitation values with a zero value.

 The elevations of the gauging stations were used when applying the KED and IED procedures for the "leave-one-out" cross validation. When interpolating temperature and precipitation exhaustively over the study area, the elevation predictors were based on the digital elevation model (DEM) of the shuffle radar topography mission (SRTM; Farr et al., 2007). SRTM originally has a resolution of about 90 m. In this study, we used the SRTM elevation model resampled to a grid with 0.5x0.5 km cells from the UTM32N coordinate reference system. This spatial resolution was judged as a good balance between computational constraints and elevation accuracy.

## 4. MODEL ASSESSMENT METHODOLOGY

The way of accounting for orographic gradients in the temperature and precipitation datasets was also assessed with respect to its ability to contribute to simulations of snow covered area and streamflow at the catchment scale using the following modelling experiment.

### 4.1. Snow accounting routine (SAR)

The selected SAR (Fig. 3a) is a modified version of CEMANEIGE proposed by Valéry et al. (2014). The original version was modified to account for: a snowfall under-catch correction factor as used in the HBV snow routine (see Beck et al., 2016), the computation of fractional snow-covered area (FSC) from a snow water equivalent (SWE) threshold, and possible integration of temperature and precipitation altitudinal gradients.

**Fig. 3** Snow accounting routine: (a) conceptual scheme and (b) associated equations (modified from Valéry et al., 2014). $P$, $R$, $T_{mean}$, $M$ and $FSC$ stand for total precipitation, rainfall, mean temperature, melt and factional snow cover, respectively.

Depending on the objectives, the model can be run in a full distributed mode or according to elevation bands. As distributed (or semi-distributed) inputs, it requires the daily liquid equivalent water depth of total precipitation ($P$) and mean daily air temperature ($T_{mean}$). In the case of a semi-distributed application at the catchment scale, the first step is to divide the catchment into elevation zones of equal area. Mean areal inputs ($P$ and $T_{mean}$) are then extracted for each elevation zone from gridded temperature and precipitation datasets. In the present study, the number of elevation zones was set at five due to computational constraints and because preliminary tests showed no significant improvement in the snow-hydrological simulations when a higher spatial resolution (more elevation bands or full distribution) was used.

In each elevation band, the functions of the SAR described in Figure 3b are applied with a unique set of parameters. Internal states (snowpack represented according to snow water equivalent (*SWE*) and its thermal state *STS*) vary independently in each elevation zone according to the differences in input values. When gridded temperature and precipitation datasets interpolated without elevation dependency are used, the SAR enables forcing data for each elevation zone to be modified based on two orographic gradients (*TLR* and *PLR*) used as key parameters:

$$T_i(t) = T_i^{IDW}(t) + \left[\frac{\left(TLR + \frac{1}{2}TLR \times Si \times CSV\right)}{100} \times (y_i - y_i^{IDW}(t))\right]$$

(9)

with:

$$Si \begin{cases} \sin\left(\frac{2\pi \times (d - 80.5)}{366}\right), & lat > 0 \\ -\sin\left(\frac{2\pi \times (d - 80.5)}{366}\right), & lat < 0 \end{cases}$$

$$P_i(t) = P_i^{IDW}(t) \times \left[1 + \frac{PLR}{1000} \times (y_i - y_i^{IDW}(t))\right] \tag{10}$$

where $T_i^{IDW}(t)$ and $P_i^{IDW}(t)$ are, respectively, the mean areal temperature and precipitation interpolated based on the IDW procedure in elevation zone $i$ at time step $t$; $y_i^{IDW}(t)$ is the mean areal elevation interpolated based on the IDW procedure in elevation zone $i$ from the available gauges at time step $t$; $y_i$ is the mean areal elevation extracted from the DEM in elevation zone $i$; $TLR$ and $PLR$ are the constant temperature and precipitation lapse rates to be calibrated; $CSV$ is a coefficient of seasonal variation due to solar radiation to be applied to $TLR$ (when set to 0, no seasonal variation is applied); $Si$ is an index of seasonal change in solar radiation accounting for daytime length and ranges from -1 on the 21st of December (winter solstice) to 1 on the 21st of June (summer solstice) in non-leap years in the Northern Hemisphere ($lat > 0$), $d$ is the number of days since the 1st of January of the current year.

In the original version of CEMANEIGE, fractional snow-covered area (FSC) is calculated as follows:

$$FSC_i(t) = \min\left(\frac{SWE_i(t)}{SWE_{th}}, 1\right) \tag{11}$$

where $SWE$ is the quantity of snow accumulated in snow water equivalent (a state variable of the model, in mm), and $SWE_{th}$ is the model's melting threshold. $SWE_{th}$ is calculated as being equal to 90% of mean annual solid precipitation on the catchment considered (Valéry et al., 2014). Alternative approaches have been proposed to account for the hysteresis that exists between $FSC$ and $SWE$ during the accumulation and melt phases (Riboust et al., 2019). However, introducing such a hysteresis adds two additional free parameters to the SAR. Instead, $SWE_{th}$ was fixed to 40 mm since preliminary sensitivity analyses showed that this value gave satisfactory $FSC$ values when compared to the MODIS observations in the studied catchments.

To ensure insightful comparison with the modelling experiment, the SAR was calibrated according to different modes and degrees of freedom (Table 2). In mode *M1*, elevation dependency, which was accounted for (or not) in the $T$ and $P$ inputs based on the interpolation procedures, was tested by calibrating five parameters ($T_S$, $T_R$, $SFCC$, $\theta$, $Kf$) which control snow accumulation and melt. This mode is usually used to allow snow processes to be adjusted to local conditions and/or the errors in the $T$ and $P$ inputs. In mode *M2*, all parameters of the SAR were fixed in order to introduce two parameters ($TLR$ and $PLR$) as orographic gradients. The aim of using this mode was to account for elevation dependency in the $T$ and $P$ inputs from constant, calibrated orographic gradients while fixing the parameters that control snow accumulation and melt to physical or general values: precipitation phase determined based on a linear separation between -1 °C and +3 °C (USACE, 1956), temperature threshold for snowmelt fixed at 0 °C, degree-day melt factor set at 5 mm. °C$^{-1}$.d$^{-1}$ (mean general value taken from Hock, 2003). In mode *M3*, the same approach was chosen but an additional parameter ($CSV$) was associated with the $TLR$ gradient in order to test the value of introducing a seasonal variation in the temperature lapse rates (see Eq. 9). In mode *M4*, elevation dependency in the $T$ and $P$ inputs was also accounted for based on three ($TLR$, $CSV$ and $PLR$) parameters and two other parameters ($\theta$, $Kf$) were calibrated in addition to allow for snowmelt adjustment.

In each altitudinal band, five outputs (rainfall, snowfall, snowmelt, potential evapotranspiration and fractional snow-covered area) are computed at each daily time step. Rainfall ($R$) and snowmelt ($M$) are summed to compute the total quantity

of water available for production and transfer in the catchment. Potential evapotranspiration (*PE*) is computed for each altitudinal band using the temperature-based formulation proposed by Oudin et al. (2005):

$$PE_i(t) = \frac{R_e}{\lambda \rho} \frac{T_i(t) + 5}{100} \qquad \text{if } (T_i(t) + 5) > 0; \quad \text{else } PE_i(t) = 0 \tag{12}$$

where $R_e$ is the extra-terrestrial solar radiation (MJ.m$^{-2}$.day$^{-1}$) which depends on the latitude of the basin and the Julian day of the year, $\lambda$ is the net latent heat flux (fixed at 2.45 MJ.kg$^{-1}$), $\rho$ is the water density (set at 11.6 kg.m$^{-3}$) and $T_i(t)$ is the air temperature (°C) estimated in the elevation zone *i* at time step *t*.

**Table 2** Parameters of the snow accounting routine and their associated fixed values or ranges tested in each modelling experiment.

| Param. | Meaning | Unit | Fixed values or ranges tested | | | |
|--------|---------|------|------|------|------|------|
| | | | *M1* | *M2* | *M3* | *M4* |
| $T_S$ | Temperature between the solid and liquid phase | °C | [-3; 3] | -1 | -1 | -1 |
| $T_R$ | Thermal range for the phase separation above $T_S$ | °C | [0; 10] | 4 | 4 | 4 |
| *SFCC* | Snowfall gauge under-catch correction factor | - | [1; 3] | 1 | 1 | 1 |
| $\theta$ | Weighting coefficient for snowpack thermal state | - | [0; 1] | 0 | 0 | [0; 1] |
| $T_M$ | Temperature threshold for snowmelt | °C | $T_S + 1$ | 0 | 0 | 0 |
| *Kf* | Degree-day melt factor | mm.°C$^{-1}$.d$^{-1}$ | [0; 10] | 5 | 5 | [0; 10] |
| $SWE_{th}$ | Snow water equivalent threshold to compute FSC | mm | 40 | 40 | 40 | 40 |
| *TLR* | Temperature lapse rate | °C (100m)$^{-1}$ | - | [0;-1.5] | [0;-1.5] | [0;-1.5] |
| *CSV* | Coefficient of seasonal variation applied to *TLR* | - | - | 0 | [0; 1] | [0; 1] |
| *PLR* | Precipitation lapse rate | % (km)$^{-1}$ | - | [0; 200] | [0; 200] | [0; 200] |

The outputs of each band are averaged to estimate the total liquid output of the SAR and *PE* at the catchment scale in order to feed the combined hydrological models (Fig. 4).

## 4.2. Hydrological models

To avoid model-specific results, two well-known hydrological models (see structures in Fig. 4 and parameters in Table 3) were chosen in association with the SAR: the 4-parameter GR4J presented by Perrin et al. (2003) and a 9-parameter lumped version of the HBV model (Bergström, 1995; Beck et al., 2016), here referred to as HBV9 to avoid confusion with the original version.–

The two models were run at a daily time step and used in lumped mode with the SAR on top. The structure and the number of degrees of freedom differ between GR4J and HBV9, which should make results more generalizable.

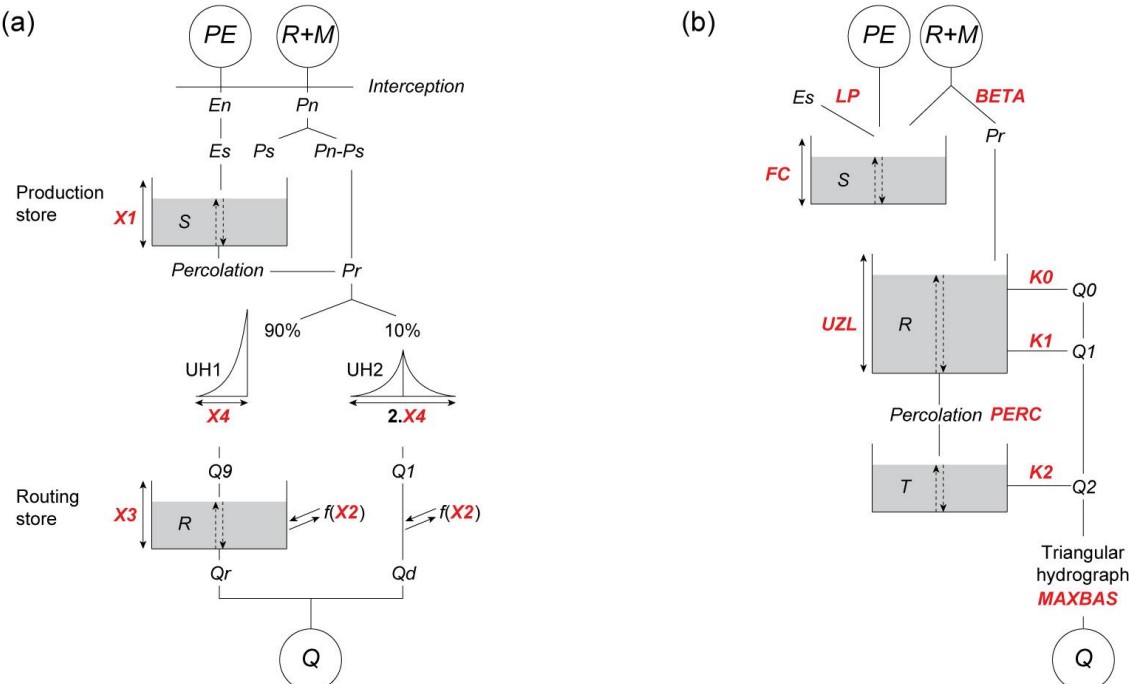

**Fig. 4** Diagrams of the two hydrological models used: (a) GR4J and (b) HBV9. Calibrated parameters are in red and are further described in Table 3. *R*, *M*, *PE* and *Q* stand for rainfall, snowmelt, potential evapotranspiration and streamflow, respectively.

**Table 3** Parameters of the hydrological models and their associated ranges tested.

| Model | Parameter | Meaning | Unit | Tested range |
|---|---|---|---|---|
| GR4J | *X1* | Maximum capacity of the production store *S* | mm | [0; 1500] |
| | *X2* | Inter-catchment exchange coefficient | mm.d$^{-1}$ | [-5; 5] |
| | *X3* | Maximum capacity of the non-linear routing store *R* | mm | [0; 500] |
| | *X4* | Unit hydrograph (UH) time base | d | [0.5; 5] |
| HBV9 | *BETA* | Shape coefficient of recharge function | - | [0.5; 5] |
| | *FC* | Maximum water storage in the unsaturated-zone store *S* | mm | [10; 1500] |
| | *LP* | Fraction of soil moisture above which actual evapotranspiration reaches PE | - | [0.3; 1] |
| | *K0* | Additional recession coefficient of the upper groundwater store *R* | - | [0.05; 1] |
| | *K1* | Recession coefficient of the upper groundwater store *R* | - | [0.1; 0.8] |
| | *UZL* | Threshold value for extra flow from the upper zone | mm | [0; 500] |
| | *PERC* | Maximum percolation to the lower zone | mm.d$^{-1}$ | [0; 6] |
| | *K2* | Recession coefficient of the lower groundwater store *T* | - | [0.01; 0.15] |
| | *MAXBAS* | Length of the equilateral triangular weighting function | - | [1; 7] |

### 4.3. Calibration and validation methods

#### 4.3.1. General model assessment

The models (GR4J and HB9 with the SAR on top) were cross-validated using a split-sample test procedure. The simulation period (2000–2016) was split into two sub-periods alternatively used for calibration and validation. Thus two calibration and two validation tests were performed to provide evaluation on all available data. Mean annual precipitation increased by around 17% between the two periods, while mean annual temperature were stable (9.2 °C vs. 9.3 °C) across the in-situ network presented in Section 2.1. Although the second period was generally wetter, differences can be observed locally. At the basin scale, the differences between the two periods ranged from -10% to +15% for precipitation, -0.5 °C to +0.5 °C for temperature, and -11% to +50% for streamflow.

The models were run in a continuous way for the whole reference period, while only hydrological years (from the 1$^{st}$ of September to the 31$^{st}$ of August) corresponding to the calibration and validation periods were considered to compute the efficiency criteria. The 3-year period before the simulation period was used for model warm-up to limit the effect of the storage initialisation, and was not included in performance computation.

### 4.3.2. Optimisation algorithm and objective function

The parameters of the SAR and the hydrological model were optimised simultaneously, using the Shuffled Complex Evolution (SCE) algorithm (Duan et al., 1992). The algorithmic parameters of SCE were set to the values recommended by Duan et al. (1994) and Kuczera (1997) to reduce the risk that SCE fails in local optimal solutions. The objective function (OF) used was a multi-criteria composite function focusing simultaneously on variations in snow-covered area and streamflow dynamics at the basin scale, as follows:

$$OF = 1 - (0.5 \times NSE_{SNOW} + 0.5 \times NSE_{sqrQ})$$

with:

$$NSE_{SNOW} = \frac{1}{E} \sum_{i=1}^{E} \left( 1 - \frac{\sum_{t=1}^{N}(FSC_{sim,t}^{i} - FSC_{obs,t}^{i})^2}{\sum_{t=1}^{N}(FSC_{obs,t}^{i} - \overline{FSC_{obs}^{i}})^2} \right) \qquad (13)$$

$$NSE_{sqrQ} = 1 - \frac{\sum_{t=1}^{N}(\sqrt{Q_{sim,t}} - \sqrt{Q_{obs,t}})^2}{\sum_{t=1}^{N}(\sqrt{Q_{obs,t}} - \overline{\sqrt{Q_{obs}}})^2}$$

where $FSC_{obs,t}^{i}$ and $FSC_{sim,t}^{i}$ are the observed and simulated fractional snow-covered area (FSC) in elevation zone $i$ at daily time step $t$, $N$ is the total number of time steps, $\overline{FSC_{obs}^{i}}$ is the mean observed FSC in elevation zone $i$ over the test period, $E$ is the total number of elevation zones (fixed to 5 for the study), $Q_{obs,t}$ and $Q_{sim,t}$ are the observed and simulated streamflows at daily time step $t$, $\overline{\sqrt{Q_{obs}}}$ is the mean observed square root transformed flows over the test period.

NSE$_{SNOW}$ relies on the Nash-Sutcliffe Efficiency criterion. Perfect agreement between the observed and simulated values gives a score of 1, whereas a negative score represents lower reproduction quality than if the simulated values had been replaced by the mean observed values. NSE$_{sqrQ}$ can be considered as a multi-purpose criterion focusing on the simulated hydrograph. It puts less weight on high flows than the standard NSE on non-transformed discharge (Oudin et al., 2006). As the majority of basins had negligible glacierized areas (see Table 1), no specific glacier model was activated. This led to ignore the late summer contribution of glacier melt to river discharge in the three basins having 9–12% glacierized areas.

### 4.3.3. Efficiency criteria in validation

Four criteria were used to evaluate model performance during validation. The first one was the NSE$_{SNOW}$ criterion. To put more emphasis on high and low flow conditions, we used the NSE on non-transformed streamflows (NSE$_Q$) that gives more weight to large errors generally associated with peak flows, and the NSE on log-transformed streamflows (NSE$_{lnQ}$). The absolute cumulated volume error (VE$_C$) was also computed to obtain information on the agreement between observed and simulated total discharge over the test periods:

$$VE_C = 1 - \frac{\left| \sum_{t=1}^{N} Q_{sim,t} - \sum_{i=1}^{N} Q_{obs,t} \right|}{\sum_{t=1}^{N} Q_{obs,t}} \qquad (14)$$

A value of 1 indicates perfect agreement while values less than 1 indicate over- or underestimation of the volume.

## 5. RESULTS AND DISCUSSION

### 5.1. Cross-validation of the interpolation methods

Table 4 lists the results of cross-validation of the interpolation methods against yearly, monthly and daily series from temperature and precipitation gauges. Kriging with ORK led to an improvement over IDW only for precipitation interpolation at the yearly and monthly time scales. Considering elevation dependency with external drift (KED and IED) improved the performance of the kriging and inverse-distance methods, except for precipitation estimated at the daily time scale. This shows that the correlation between precipitation and topography increases with the increasing time aggregation as already reported in other studies (e.g.,  Bárdossy and Pegram, 2013; Berndt and Haberlandt, 2018). The elevation-dependency of precipitation thus depends significantly on the accumulation time. At the daily time scale, the orographic enhancement is limited because on a given day there is no monotonic relationship between elevation and precipitation amount: it depends on where the precipitation event occurs in the first place.

Of all the methods tested, IED provided the best performance in terms of lower RMSE for each variable (temperature and precipitation) and at all temporal resolutions (yearly, monthly and daily), except for precipitation at the daily time scale for which IDW performed best.

The exponential variogram model for the ORK and KED method performed systematically better than the spherical model whatever the variable and the time scale.  In contrast, the exponent $\omega$ used with the IDW and IED methods varies from 1 to 3 depending on the considered variable and time scale. The optimised number of surrounding neighbours $n(u)$ also varies depending on the method and time scale. At the daily time scale, $n(u)$  ranged from 6 when interpolating temperature with ORK to 17 when interpolating precipitation with KED and IED. Hence, 10 (17) surrounding neighbours were used to compute altitudinal gradients of temperature (precipitation) based on the daily linear regressions with KED and IED.

**Table 4** Cross-validation of the interpolation methods against yearly, monthly and daily series from meteorological gauges over the period 2000–2016. The best efficiency criteria for each analytical time scale and each variable of interest (temperature and precipitation) are in bold. The values of $n(u)$ and $\omega$ (for IDW and IED) and of $n(u)$ and *model* (for ORK and KED) represent the interpolation parameters, which were optimised using the leave-one-out procedure, as described in section 3.3.

| | | Temperature (78 gauges) | | | | Precipitation (148 gauges) | | | |
|---|---|---|---|---|---|---|---|---|---|
| | | Without elevation dependency | | With elevation as external drift | | Without elevation dependency | | With elevation as external drift | |
| | | IDW | ORK | KED | IED | IDW | ORK | KED | IED |
| Yearly | RMSE | 1.82 °C | 1.82 °C | 0.65 °C | **0.65 °C** | 177.05 mm | 174.27 mm | 153.75 | **150.31 mm** |
| | $n(u)$ | 6 | 6 | 8 | 10 | 4 | 15 | 12 | 12 |
| | $\omega$ | 1 | - | - | 2 | 3 | - | - | 3 |
| | model | - | exponential | exponential | - | - | exponential | exponential | - |
| Monthly | RMSE | 1.91 °C | 1.92 °C | 0.86 °C | **0.80 °C** | 23.19 mm | 22.73 mm | 22.35 mm | **22.20 mm** |
| | $n(u)$ | 7 | 6 | 8 | 10 | 5 | 15 | 12 | 12 |
| | $\omega$ | 1 | - | - | 2 | 2 | - | - | 2 |
| | model | - | exponential | exponential | - | - | exponential | exponential | - |
| Daily | RMSE | 2.16 °C | 2.18 °C | 1.21 °C | **1.20 °C** | **2.83 mm** | 2.86 mm | 2.91 mm | 2.90 mm |
| | $n(u)$ | 7 | 6 | 10 | 10 | 10 | 10 | 17 | 17 |
| | $\omega$ | 1 | - | - | 2 | 2 | - | - | 2 |
| | model | - | exponential | exponential | - | - | exponential | exponential | - |

Figure 5 shows the annual temperature and precipitation maps obtained by interpolation daily data from the meteorological gauges with the period 2000–2016 using the IDW, ORK, KED and IED methods and their optimised parameters (Table 4). The maps of IED estimates of mean temperature and annual precipitation closely resemble the KED maps. Temperature estimates range from -9.2 °C to 16.0 °C with KED, and from -11.2 °C to 16.6 °C with IED. Precipitation estimates range from 630 mm to 3273 mm with KED, and from 642 mm to 3184 mm with IED. As expected, these ranges are wider than those obtained with the IDW and ORK procedures, which, unlike the KED and IED methods, do not consider either local or

seasonal elevation dependency. As a result, the ranges obtained with KED and IED are probably more realistic with respect to temperature, but not necessarily with respect to precipitation, for which daily cross-validation shows that the simple IDW method provided better results. However, cross-validation was based on gauges sampled only below 2006 m a.s.l. (2105 m a.s.l.), meaning evaluation of the precipitation (temperature) gridded datasets at higher altitudes was not possible. Another approach is thus needed to further explore whether elevation dependency should be disregarded when estimating daily precipitation (as suggested by cross-validation), and, if not, whether this dependency should be accounted for in the interpolation process or by inverting the hydrological cycle. A sensitivity analysis of snow-hydrological simulations to the orographic gradients was thus conducted.

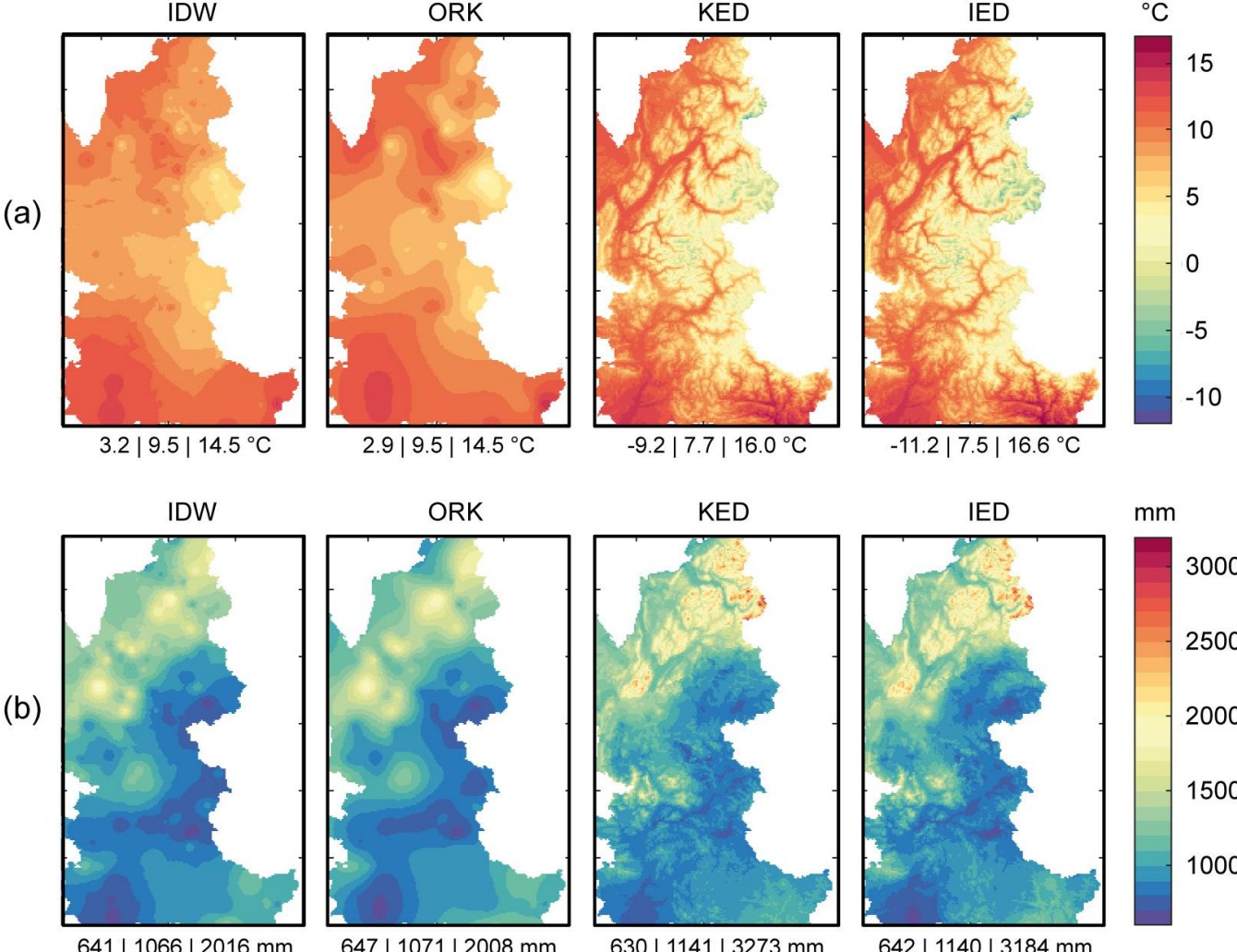

**Fig. 5** Maps of mean annual temperature (°C) and total precipitation (mm per year) obtained by interpolation of (a) 78 gauges (temperature) and of (b) 148 gauges (precipitation) with daily data for the period 2000–2016 using the inverse distance weighted (IDW), ordinary kriging (ORK), kriging with external drift (KED) and IDW with external drift (IED). The numbers below each map stand respectively for minimum, mean and maximum values (expressed in °C for temperature and in mm/year for precipitation) in the maps.

## 5.2. Sensitivity of the snow-hydrological simulations to the orographic gradients

For the sake of brevity, here we only present the results obtained with the datasets interpolated with the IDW and IED procedures, since cross-validation at the daily time scale showed that they slightly outperformed the ORK and KED methods, respectively. Table 5 summarises the six tests performed to account for elevation dependency in the $T$ and $P$ inputs via the modelling experiment described in section 4.

**Table 5** Description of the tests to account for elevation dependency in the T and P inputs via the modelling experiment described in section 4. Note that each calibration tests included also the hydrological parameters of GR4J or HBV9 (the parameter ranges tested are listed in Table 2 for the SAR and in Table 3 for the hydrological models).

| Mode | Test number | $T$ input | $P$ input | Calibrated parameters (excluding hydrological models) | Principle |
|------|------|------|------|------|------|
| *M1* | 1 | T-IDW | P-IDW | $T_S$, $T_R$, SFCC, $\theta$, Kf | No elevation dependency in the $T$ and $P$ inputs, and five calibrated parameters for adjustment of snow accumulation and melt |
| | 2 | T-IED | P-IDW | $T_S$, $T_R$, SFCC, $\theta$, Kf | Elevation dependency only in the $T$ input based on the IED interpolation procedure, and five calibrated parameters for adjustment of snow accumulation and melt |
| | 3 | T-IED | P-IED | $T_S$, $T_R$, SFCC, $\theta$, Kf | Elevation dependency in the $T$ and $P$ inputs based on the IED interpolation procedure, and five calibrated parameters for adjustment of snow accumulation and melt |
| *M2* | 4 | T-IDW | P-IDW | *TLR, PLR* | Elevation dependency in the $T$ and $P$ inputs considered based on two calibrated parameters in the SAR, and fixed parameters for snow accumulation and melt |
| *M3* | 5 | T-IDW | P-IDW | *TLR, CSV, PLR* | Elevation dependency in the $T$ and $P$ inputs considered based on three calibrated parameters in the SAR, and fixed parameters for snow accumulation and melt |
| *M4* | 6 | T-IDW | P-IDW | *TLR, CSV, PLR*, $\theta$, Kf | Elevation dependency in the $T$ and $P$ inputs considered based on three calibrated parameters in SAR, and two calibrated parameters for adjustment of snow melt |

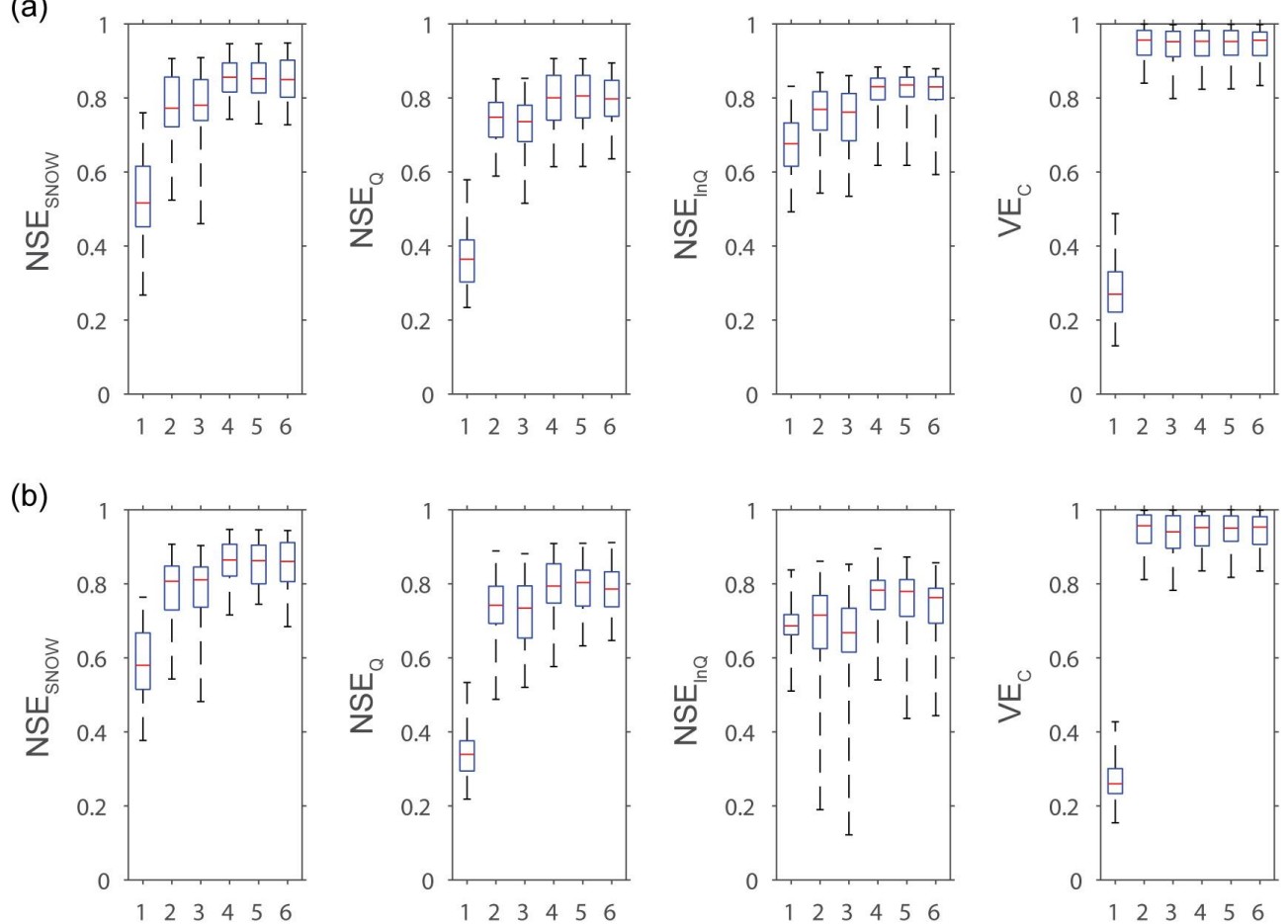


**Fig. 6** Boxplots (showing 0.00, 0.25, 0.50, 0.75 and 1.00 percentiles) of the efficiency distributions obtained in validation by the (a) GR4J and (b) HBV9 models combined with the snow model according to six different tests (see Table 5) to account for elevation dependency in the $T$ and $P$ inputs on the 20 snow-affected Alpine catchments.

Figure 6 and Table 6 summarise the efficiency distributions obtained in validation with the GR4J and HBV9 models combined with the snow model in the different tests on the 20 snow-affected Alpine catchments. The results produced by the two hydrological models are in agreement and highlight the following main findings:

- − Not considering elevation dependency in either the $T$ or $P$ inputs (Test #1) leads to notable failures of the snow-hydrological models, due to incorrect snow/rainfall partitioning and snowmelt in space and over time caused by too

high temperatures and insufficient input volumes of precipitation, which cannot be offset by the free parameters of the SAR. Notably, calibrating temperature thresholds and ranges for snow/rain partition and snow melt, as well as snow under-catch (using the *SFCC* parameter) is clearly unsatisfactory.

– Considering elevation dependency only in the *T* inputs based on the IED procedure (Test #2) significantly improves the snow-hydrological simulations, but considering elevation dependency in the *P* inputs based on the same
procedure (Test #3) is not as efficient, notably for streamflow simulations. This shows that the estimated precipitation with IED over the catchments is of limited accuracy.

– Improving the areal temperature and precipitation estimation clearly requires the calibration of altitudinal temperature and precipitation gradients. The snow-hydrological simulations are considerably improved when using the parsimonious 2-parameter SAR based only on the calibration of *TLR* and *PLR* (Test #4).

– Compared to Test #4 based only on a 2-parameter SAR, only limited improvements in the performance distributions are obtained by introducing additional free parameters to account for the seasonal variability of the temperature gradients (Test #5) and for local adjustment of snowmelt (Test #6).

**Table 6** Mean validation efficiency of the 6 modelling tests (see Table 5) on the set of 20 catchments with the GR4J model and the HBV9 model.

| Model | Test number | Number of free parameters of the SAR | Mean $NSE_{SNOW}$ | Mean $NSE_Q$ | Mean $NSE_{lnQ}$ | Mean $VE_C$ |
|---|---|---|---|---|---|---|
| GR4J | 1 | 5 | 0.53 | 0.37 | 0.66 | 0.29 |
|  | 2 | 5 | 0.78 | 0.74 | 0.76 | 0.94 |
|  | 3 | 5 | 0.79 | 0.72 | 0.74 | 0.94 |
|  | 4 | 2 | 0.86 | 0.79 | 0.82 | 0.95 |
|  | 5 | 3 | 0.85 | 0.80 | 0.82 | 0.95 |
|  | 6 | 5 | 0.85 | 0.80 | 0.81 | 0.95 |
| HBV9 | 1 | 5 | 0.59 | 0.34 | 0.69 | 0.27 |
|  | 2 | 5 | 0.79 | 0.74 | 0.68 | 0.94 |
|  | 3 | 5 | 0.78 | 0.72 | 0.65 | 0.93 |
|  | 4 | 2 | 0.86 | 0.79 | 0.76 | 0.94 |
|  | 5 | 3 | 0.86 | 0.79 | 0.75 | 0.94 |
|  | 6 | 5 | 0.85 | 0.79 | 0.75 | 0.94 |

As a representative example of the studied catchments, Figure 7 illustrates the differences in the simulations obtained by Tests #1 to #4 for the Durance at Serre-Ponçon. This 3580 km² catchment with altitudes ranging between 652 and 4017 m a.s.l. ensures inflows to one of the biggest dams in Europe (maximum capacity of 1.3 km³). Dynamics of fractional snow
cover area and streamflow are better simulated when considering elevation dependency of the *T* and *P* inputs via two calibrated, altitudinal gradients (Test #4). Compared to the other tests, mean annual temperature (4.0 °C) is lower and mean annual precipitation (1160 mm) is higher. Less precipitation is considered in solid form (44% of total precipitation on average) and accumulation is longer during winter, which fits streamflow observations better, both for low flows from December to April and for flood peaks between May and July. It is worth noting that these improved simulations were
obtained with a SAR calibrated on only two parameters targeting the local lapse rates whereas the other simulations were based on a SAR calibrated on five parameters. This shows that calibrating the usual snow parameters to compensate for errors in the input data and/or to adapt to local snow-related processes is less efficient in the simulations than inferring only temperature and precipitation lapse rates while fixing all the other parameters. This suggests that correcting for temperature and precipitation distribution has a stronger impact on model predictions than adjusting for snow-related processes like phase
partitioning or melt and that correctly estimating total accumulation is likely to play a first-order role in the snow-hydrological responses of the studied catchments.

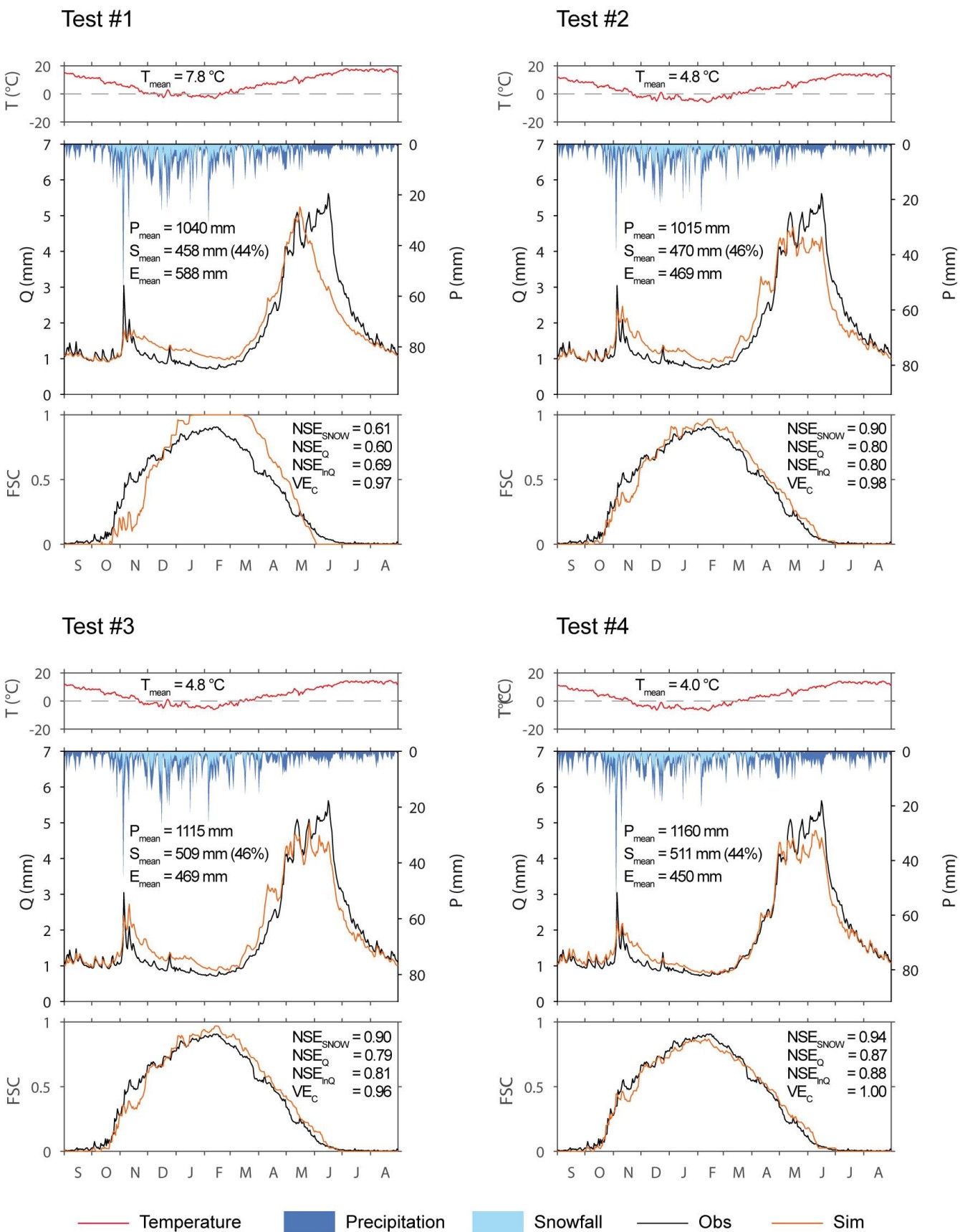

**Fig. 7** Comparison of snow-hydrological simulations with elevation dependency according to Tests #1 to #4 (see Table 5) with GR4J for the Durance at Serre-Ponçon. The graphs show mean inter-annual time-series of temperature, precipitation, streamflow and fractional snow cover at the catchment scale in validation over the period 2008–2016. $T_{mean}$, $P_{mean}$ and $S_{mean}$ stand for mean annual temperature, precipitation, and snowfall, respectively. The efficiency criterions $NSE_{SNOW}$, $NSE_Q$, $NSE_{lnQ}$ and $VE_C$ are computed from continuous (not mean seasonal) series over 2008–2016.

## 5.3. Identifiability of the parameters

Figure 8 shows a representative example of parameter sensitivity to the objective function (*OF*) according to the six tests (see Table 5) with the GR4J model on the Durance at Serre-Ponçon. The maximum allowed parameter range is only reached for the parameters *X1* and *X2* with Test #1. This test differs from the others because no elevation dependency in the *T* and *P* inputs are considered. Consequently, hydrologic predictions of Test #1 are significantly outperformed by the other approaches. Extending the parameter ranges beyond the tested values would be both poorly efficient in improving the

simulations and incorrect from a numerical point of view since they were set to values recommended by the models' authors. Moreover, no maximum parameter limits were reached in the other tests, thus suggesting that the parameter ranges are adequate.

    As already shown, considering elevation gradients (Tests #4, #5 and #6) minimises OF and significantly improves model performance. It also improves the parameter identifiability. The temperature altitudinal gradient (*TLR*) is easily

identifiable with values ranging from -0.64 °C/100m (Test #4 and Test #5) to -0.67 °C/100m (Test #6), and with variation coefficients of 0.1% for the 20% best-performing parameter solutions. It clearly reveals as a key parameter for improving snow and streamflow simulations compared to parameters calibrated using elevation gradients inferred from usual interpolation method with external drift (Tests #2 and #3). The optimum value of the *CSV* parameter (Tests #5 and #6) is zero, clearly indicating no need to account for the seasonal variation in the temperature lapse rate in the catchment studied

here (which is also the case in most catchments). The precipitation lapse rate (*PLR*) is also easy to identify with optimised values of 62%/km (Tests #4–5) and variation coefficients 0.4%. Introducing additional parameters controlling snowmelt ($\theta$ and *Kf* in Test #6) does not significantly improve the simulations and decreases the parameter identifiability (variation coefficients increase compared to Tests #4 and Test #5 based on a 2-parameter and 3-parameter SAR, respectively). This shows that model performance is mainly sensitive to the use of parameters for temperature and precipitation lapse rates and

that a 2-parameter SAR based on *TLR* and *PLR* (Test #4) on top of the hydrological models tested is both essential and sufficient to achieve satisfactory simulations.

    Equifinality is also reduced in Tests #4–6 for the parameters controlling runoff generation and routing (*X1*, *X3* and *X4*). On the opposite, the parameter of the inter-catchment groundwater flows (*X2*) is poorly identifiable with variation coefficients of 24.8%, 20.3% and 143.1% with Test #4, Test #5 and Test #6, respectively. This suggests that inter-catchment

groundwater exchanges (IGE) do not play a key role in the studied catchments. Indeed, fixing *X2* to a value of 0 (i.e. without potential IGE) with an alternative GR3J model provided similar mean validation efficiency on the set of catchments as compared to the GR4J associated with the 2-parameter SAR (Table 7). However, other objective functions may result in other findings as far as IGE are concerned. For instance, additional tests (not shown here for brevity sake) confirmed that it was possible to greatly reduce the *X2* equifinality without decreasing the model efficiency by adding a water balance term in

the objective function to constrain the proportion of years respecting the water and energy balance in the Turc-Budyko non-dimensional graph (see Andréassian and Perrin, 2012). These tests suggested that it may be relevant to explicitly represent inter-catchment groundwater transfers in association with correcting or scaling factors applied to the precipitation input data to render the distribution between evapotranspiration, streamflow and underground fluxes more realistic, as already reported by Le Moine et al. (2007).


**Table 7** Mean validation efficiency on the set of 20 catchments with the GR4J model and the GR3J model in association with the 2-parameter SAR.

| Model | Total number of free parameters | Mean NSE$_{SNOW}$ | Mean NSE$_Q$ | Mean NSE$_{lnQ}$ | Mean VE$_C$ |
|---|---|---|---|---|---|
| 2-parameter SAR/GR4J | 6 (2 + 4) | 0.86 | 0.79 | 0.82 | 0.95 |
| 2-parameter SAR/GR3J | 5 (2 + 3) | 0.86 | 0.78 | 0.81 | 0.94 |

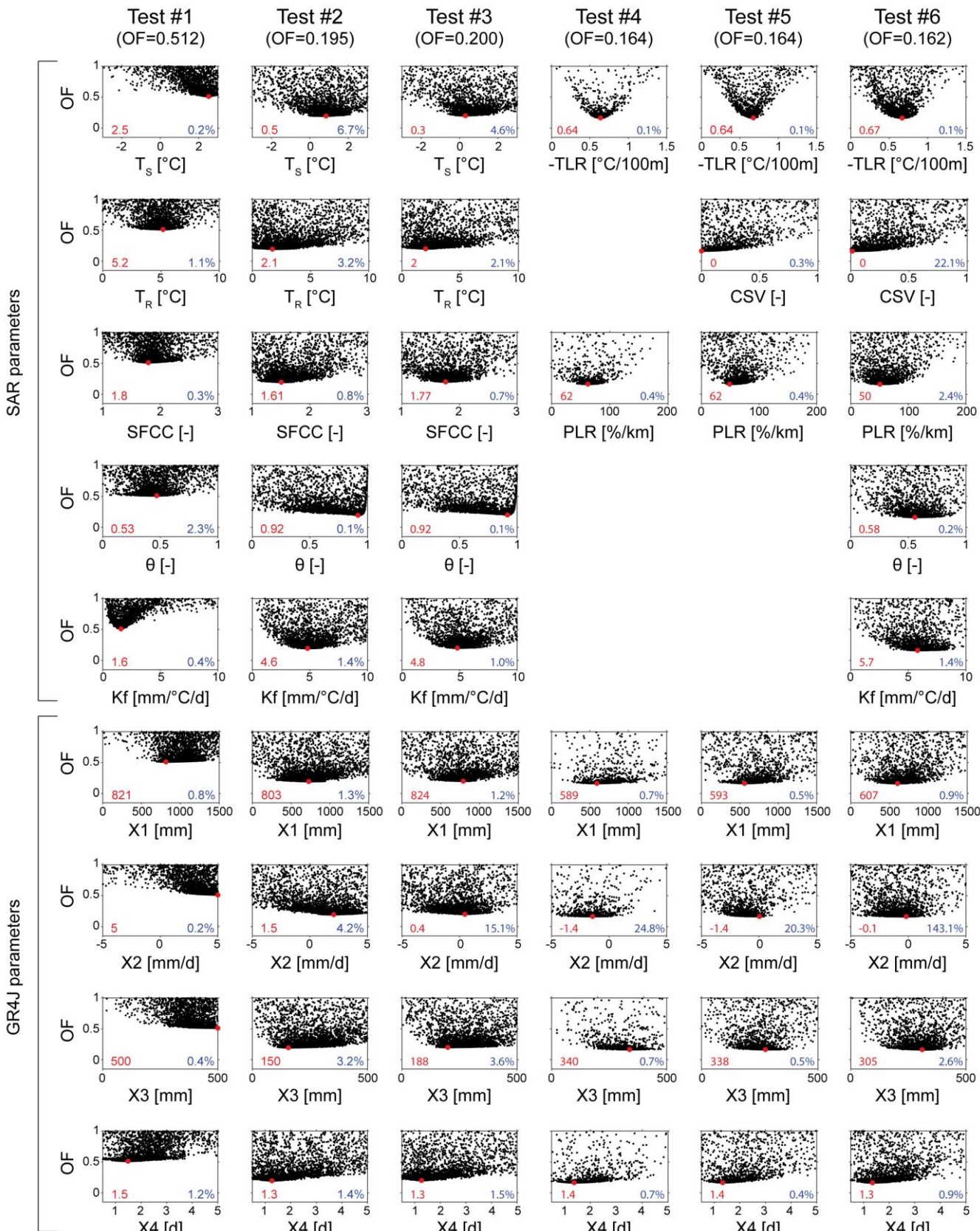

**Fig. 8** Parameter sensitivity to the objective function (OF) according to Tests #1 to #6 (see Table 5) with GR4J combined with the snow accounting routine (SAR) on the Durance at Serre-Ponçon. The values and dots in red indicate the optimised calibrated parameters when minimising OF, the black dots represent trials of the SCE-UA optimisation algorithm, and the values in blue are the variation coefficients (in %) of the 20% best-performing parameter solutions compared to the optimised values for each parameter (the lowest value, the easiest parameter identifiability). Note that depending on the tests, the calibrated parameters of the SAR vary from 2 to 5 (see Table 5 and Table 2), while the GR4J hydrological models has 4 free parameters (see Table 3).



### 5.4. Ranges of the calibrated altitudinal gradients

Figure 9 shows that the temperature and precipitation lapse rates vary considerably from one catchment to another.

The mean value of the calibrated temperature lapse rates is -0.68 °C (100m)[-1] and -0.65 °C (100m)[-1] with GR4J and HBV9, respectively. These values are higher than the yearly lapse rates identified by Rolland (2003) from gauge observations in Alpine regions, which ranged from -0.54 to -0.58 °C (100m)[-1] in the Italian and Austrian Tyrol. Instead, the mean calibrated values are close to the average temperature gradients generally proposed as approximations in the literature (-0.65 °C (100m)[-1] in Barry and Chorley, 1987). They can be used as suitable estimates for daily snow-hydrological purposes in the French Alps. However, to better account for local meteorological conditions, it may be advisable to calibrate them

since the *TLR* parameter ranges from -0.41 to -0.83 °C (100m)[-1] depending on the catchments and on the models, and is easily identifiable (see Fig. 8 and section 5.3.).

The mean value of the calibrated precipitation lapse rates is 30% (km)[-1] and 29% (km)[-1] with GR4J and HBV9, respectively. The differences in ranges between the two models may be due to the GR4J ability to gain (or loose) water from inter-catchment groundwater flows through its *X2* parameter (see section 5.3.), unlike HBV9 which considers the catchment

as a closed system. On the other hand, HBV9 relies on more parameters for production and transfer, thus enabling to compensate differently for the errors in the precipitation volumes. Whatever the model, the calibrated lapse rates indicate the need for increased precipitation volumes in most catchments, either to counterbalance for erroneous measurements such as the systematic errors associated with precipitation under-catch during snowfall, or to consider the orographic effect that cannot be sufficiently accounted for by the gauges used for interpolating the precipitation fields. However the ranges of the

precipitation lapse rates, from 0 to 100% (km)[-1] with GR4J and from 0 to 82% (km)[-1] with HBV9, suggest that the required correction is catchment-specific and depends either on the local meteorological conditions or on data from the available surrounding stations to interpolate the daily precipitation.

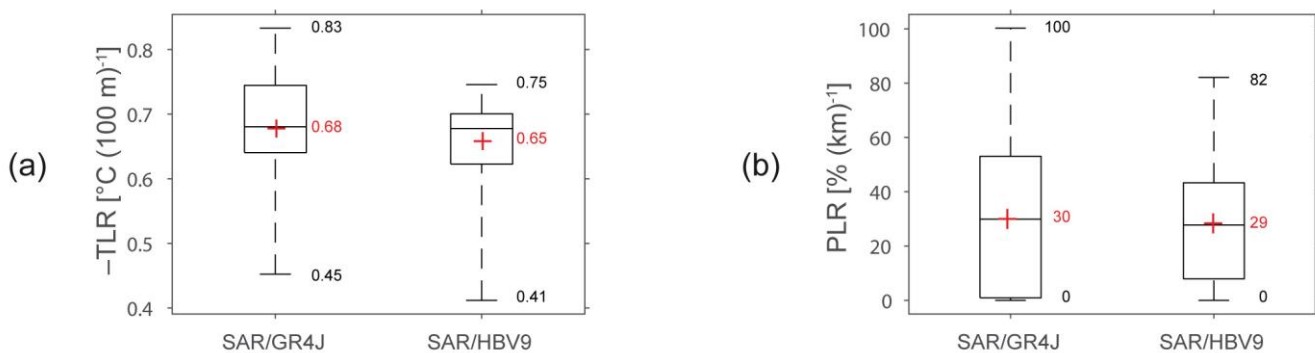

**Fig. 9** Boxplots (showing 0.00, 0.25, 0.50, 0.75 and 1.00 percentiles) of the ranges of (a) temperature and (b) precipitation lapse rates calibrated with the 2-parameter SAR (Test #4) in association with the GR4J and HBV9 models on the 20 snow-affected Alpine catchments. The red crosses indicate mean values.

## 6. SUMMARY AND CONCLUSIONS

### 6.1. Synthesis

In this paper, several alternative approaches for distributing daily temperature and precipitation are compared in the French Alps.. Elevation dependency in the temperature and precipitation fields was accounted for using two main strategies: (1) by estimating the local and time-varying altitudinal gradients from the available gauge network based on deterministic (inverse distance weighted) and geostatistical (kriging) methods with external drift; and (2) by calibrating the local gradients using an inverse snow-hydrological modelling procedure.

Cross-validation of the mapping methods showed that, whatever the time scale, temperature estimates can clearly benefit from taking altitude into account with interpolation methods based on external drift. For precipitation, incorporating

elevation in the interpolation methods was helpful for yearly and monthly accumulation times but could not achieve an improvement for daily time resolution. Results also showed that accounting for elevation dependency from gauge networks when interpolating air temperature and precipitation was not sufficient to provide accurate inputs for the snow-hydrological models tested here. The lack of high-elevation stations seriously limited correct estimation of local, time-varying lapse rates of temperature and precipitation, which, in turn, affected the performance of the snow-hydrological simulations due to too imprecise estimates of temperatures and of precipitation volumes. Conversely, optimizing lapse rates as part of a snow-hydrological modelling procedure provided evidence for increased accuracy in the simulation of snow cover and discharge dynamics, while limiting the problems of over-calibration and equifinality through parsimonious parametrisation.

## 6.2. Recommendations

These results suggest that interpolation methods using elevation as external drift such as those tested (KED and IED) should be used with caution in the absence of sufficient high-elevation data. Although the gauge density in the French Alps is close to the minimum density recommended by WMO (2008) for mountainous areas, the number of weather stations is insufficient for a complete cover of the altitude ranges. This seriously limits estimates of local and seasonal relations with elevation, notably for daily precipitation, but also for temperature, which was initially not apparent when using the leave-one-out procedure against available gauges. Placing meteorological fields in a snow-hydrological perspective thus proved indispensable to confirm the limited suitability of standard interpolation methods for generating reliable spatially distributed modelling inputs in mountainous areas. It also made it possible to propose a modelling approach to correct meteorological inputs in complex, mountainous environments and showed that it is possible (and even advisable) to use remotely-sensed snow-cover and streamflow measurements to improve our knowledge of temperature and precipitation inputs in data-scarce mountainous regions. Using auxiliary observations of snow cover notably proved to be useful to give additional insights into the reliability of the modelled snow processes.

However the differences in the two compared approaches are worth discussing. The first is to regionalize temperature and precipitation based on in-situ data and various interpolation/extrapolation schemes based on IDW or Kriging; the second is to "embed" part of the distribution process into the snow-hydrologic models via calibrated lapse rates correcting a first-guess distribution based on IDW. While the first approach is independent from hydrological data like fractional snow cover and streamflow, the second does take advantage of these data to adjust some of the distribution parameters. The second strategy prove superior to the first, especially since calibrating distribution parameters rather than adjusting snow parameters allowed the models to significantly improve their performance. This improvement was however assessed based on the same hydrological variables that were used to calibrate the snow-hydrologic models, rather than on independent measurements of temperature and precipitation. This left wondering if improving hydrologic predictions by calibrating the local gradients using an inverse snow-hydrological modelling framework also improves actual temperature and precipitation estimates. In principle, one would expect the obtained altitudinal gradients to be both more effective in terms of hydrologic predictions and in terms of temperature and precipitation, but the improvement obtained by "embedding" part of the distribution process into the snow-hydrologic models was only quantified in terms of modelling skills. An improved fit for hydrologic variables may not automatically mean that the model is also better representing weather patterns of temperature and precipitation. A good example is that the optimized lapse rates (Fig. 9) can locally be different between the two hydrologic models considered. Since independent data of temperature and precipitation at high elevations are not available, it was not possible to clarify the extent to which these results apply to temperature and precipitation in addition to hydrologic variables. As pointed out by Dettinger (2014), we are largely blind to what is happening in these high-altitude regions.

Another key-issue concerns the level of complexity required to control snow accumulation and melt. Most degree-day snow models in the literature use free parameters to adjust snowpack processes and streamflow responses, including the

whole water balance. Some parameters (temperature thresholds for snow/rain partition and snowmelt, solid precipitation correction factor) aim to compensate for the errors in the $T$ and $P$ inputs, while others (thermal state of the snowpack, degree-day melt factor) aim to fit snowmelt to local conditions. Our results showed that calibrating these parameters based on a 5-parameter SAR was much less efficient in improving the modelling performance than fixing them and calibrating only local temperature and precipitation altitudinal gradients based on a simple 2-parameter SAR. These results show that altitudinal gradients of temperature and precipitation inputs should be inferred from key parameters in snow-hydrological models since they play a first-order role in snow-hydrological simulations. Accurate estimate of these parameters greatly helps in determining the form of precipitation and spatial distribution of temperature and precipitation, and are critical for snow cover and runoff modelling in high mountain catchments, as already reported in other regions (Zhang et al., 2013; Naseer et al., 2019). Inferring the gradients reduces the input errors originating from the non-representative vertical distribution of stations while allowing the parameters of snow accumulation and melt to be set at general or physical values (see Table 2 in Section 4.1). Introducing additional free parameters to account for the seasonal variability of the temperature gradients and for adjustment of snowmelt led to only limited improvements in the performance distributions compared to the simulations based on the parsimonious 2-parameter SAR. This finding suggests that correcting errors in the model inputs is more critical than adapting the SAR to local snow processes. It also suggests limiting the degree of freedom allowed in degree-day snow models in order to reduce the risk of over-parametrisation.

## 6.3. Prospects

It would be instructive to further explore the sensitivity of snow-hydrological simulations to seasonal variations in the lapse rates, e.g. by using daily altitudinal gradients instead of a uniform constant gradient at the basin scale. However, as shown in the present paper, establishing the relationship between temperature/precipitation and elevation at the daily time scale from a sparse network of gauges is challenging in mountainous regions. For temperature, the methods tested for computing local and daily lapse rates for each prediction point (KED, IED) outperformed the methods that did not account for altitudinal gradients (IDW, ORK) in the leave-one-out procedure. On the other hand, using only a constant lapse rate calibrated from the inverse modelling approach performed substantially better as regards to the snow-hydrological predictions than using the interpolated datasets of temperature with external drift. This shows either that the local temperature lapse rates (including their seasonal variation) were not correctly captured by the daily application of interpolation methods with external drift, or that, for our experiment, accurately estimating a constant, uniform gradient for temperature was more important than estimating its seasonal variations for hydrological simulations. A seasonal variation in temperature gradient was also tested with a sinusoidal approach, which required an additional free parameter to determine the variation interval. When we compared the modified 3-parameter SAR version (Test #5 in Table 5) with the 2-parameter SAR, we found that the snow-hydrological performance distributions of the two SARs were very similar (Fig. 6). This means that, although the seasonal variation in the temperature altitudinal gradient can be put in evidence from gauge networks, as shown by Rolland (2003) for alpine regions, it did not appear indispensable for the daily snow-hydrological processes represented in our modelling experiment. Alternatively, improving the snow-hydrological simulations could consist in using minimum and maximum air temperature rather than daily mean temperature (see e.g. Turcotte et al., 2007) in order to better determine the snow/rain partition. For the regionalisation of these extreme temperatures, one challenge that remains will be characterising the high variability of daily lapse rates, which reflects temperature inversions as well as rapidly changing circulation patterns, as reported in Stahl et al. (2006). The problem is even more challenging for precipitation whose lapse rates could not be related to seasonal or other types of systematic variations as they are strongly dependent on the synoptic meteorological conditions and therefore highly variable. Further research could thus build on the works of Jarvis and Stuart (2001) for temperature and

Gottardi et al. (2012) for precipitation and focus on methods for interpolation and extrapolation that are capable of accounting for differences in the influence of topography in different seasons and synoptic situations.

Finally, it is worth mentioning that spatial variability was only considered along five elevation bands in each catchment since preliminary tests showed no improvement in the hydrologic predictions when applying the SAR in a full distribution mode. However, the SAR was not designed to explicitly account for topographic effects (slope, aspect and shading) on snow accumulation, redistribution and melt (see e.g. Frey and Holzmann, 2015). For instance, a grid-based temperature-index model could be implemented to include potential clear-sky direct solar radiation at the surface, thus considering both the seasonal variations of melt rates and the geometric effects on melt attributable to terrain (see e.g. Hock, 1999). It would thus be interesting to assess whether accounting for the influence of such effects can further improve the daily hydrologic predictions at the basin scale.

**Data availability.** The hydro-meteorological data and MODIS snow products used in this study are available via the respective websites of the dataset producers: Météo-France (https://publitheque.meteo.fr), Banque Hydro (http://www.hydro.eaufrance.fr), and NASA's National Snow and Ice Data Center (NSIDC, https://nsidc.org) Distributed Active Archive Center (DAAC).

**Author contribution.** Denis Ruelland conceived the study, performed the analysis and wrote the paper.

**Competing interest.** The author declares that he has no known competing financial interests or personal relationships that could have appeared to influence the work reported in this paper.

**Acknowledgements** The author is very grateful to *Météo-France* (https://publitheque.meteo.fr) and *Banque Hydro* (http://www.hydro.eaufrance.fr) for providing the necessary public hydro-meteorological data for the study. NASA's National Snow and Ice Data Center (NSIDC) Distributed Active Archive Center (DAAC) is also acknowledged for providing MODIS snow products (https://nsidc.org), which were used for model calibration and validation. Finally, the author is sincerely grateful to the two anonymous reviewers for they careful reading of the original manuscript and their insightful comments and constructive suggestions for improvements.

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
