# Peer review of "Should altitudinal gradients of temperature and precipitation inputs be inferred from key parameters in snow-hydrological models?"

_Hydrology and Earth System Sciences, 2019_

## Referee Comment (RC1) · Anonymous Referee #1 · 27 Dec 2019

This paper presents a nicely conceived study where several alternative approaches for distributing temperature and precipitation are compared in a mountain region (French Alps). In addition to standard interpolation approaches based on inverse-distance weighting and Kriging, the author explores the possibility of optimizing lapse rates as part of a snow-hydrologic-model calibration procedure. Results of a split-sample test show that the latter approach provides improved results for the target variables considered during calibration, that is, fractional snow cover (FSC) from MODIS, streamflow, and the water balance. Also, optimizing the temperature and precipitation distribution algorithm with hydrologic data results in the temperature and precipitation fields being colder and wetter than those obtained by using only in-situ measurements of temper-

ature and precipitation, respectively; this result agrees with expectations, especially since the considered ground-based network is not representative of high elevations in the study catchments.

I enjoyed reading the manuscript and I think it represents an interesting contribution for HESS, especially because the investigated topic is a clear open issue in mountain hydrology. I do have several general and specific comments, which I attach below. Overall, I think that the revision is feasible.

GENERAL COMMENTS

1. in my understanding, the author essentially compares two main strategies: the first is to regionalize temperature and precipitation based on in-situ data and various interpolation/extrapolation schemes based on IDW or Kriging (Section 3); the second is to ''embed'' part of the distribution process into the snow-hydrologic models via lapse rates that will correct a first-guess distribution based on IDW (see Section 4, Table 2 and 5 and Equations 9 and 10). While the first approach is independent from hydrologic data like fractional snow cover and streamflow, the second does take advantage of these data to adjust some of the distribution parameters. In my understanding, the main point of the paper is that the second strategy is superior to the first, especially since adjusting snow parameters rather than precipitation-distribution parameters does not allow the model to significantly improve its performance (see Table 5 and Fig. 5). Unless I am missing something here, this improvement was however assessed based on the same hydrologic variables that were used to calibrate the snow-hydrologic models, rather than on independent measurements of the two variables of interest: temperature and precipitation. This left me wondering if this experiment shows that "calibrating the local gradients using an inverse snow-hydrological modelling framework" improves actual temperature and precipitation estimates, or if it shows that it improves hydrologic predictions. In principle, one would expect the obtained altitudinal gradients to be both more effective in terms of hydrologic predictions and in terms of temperature and precipitation, but the improvement obtained by ''embedding'' part of the distribu-

tion process into the snow-hydrologic models is quantified in terms of modeling skills for fractional snow cover, streamflow, and the water balance (Figs. 5 to 8) rather than for independent estimates of temperature and precipitation. If independent data of temperature and precipitation at high elevations are not available, then I would recommend the author to clarify the extent to which these results apply to temperature and precipitation in addition to hydrologic variables.

2. The point above is particularly important since hydrologic models may suffer from several sources of conceptual and parametric uncertainties, some of which are visible in the interesting Figure 8. It follows that an improved fit for hydrologic variables may not automatically mean that the model is also better representing weather patterns of temperature and precipitation. A good example here is that the obtained lapse rates (Fig. 9) can locally be quite different between the two hydrologic models considered. To me, this may challenge the idea that this approach could be used to "infer local altitudinal gradients from a sparse network of gauges based on key parameters in the snow-hydrological models" (L 592ff). It does suggest that the method improves hydrologic predictions, but implications for actual temperature and precipitation are more elusive to me and should be discussed more extensively.

3. Related to this, both hydrologic models were used in lumped mode (L340), even if several other modeling approaches explicitly account for spatial variability in hydrologic processes (e.g., raster-based models). At least some discussion on this point would be interesting.

4. Spatial variability was considered in the snow model, which was implemented along five elevation bands in each catchment. This model does include all fundamental snow processes, but in my understanding does not include a specific provision for wind drift. Relying on FSC from MODIS may sometimes lead to confounding effects in this regard, where wind-driven accumulation and erosion is mistakenly assumed as due to precipitation or melt. Was this somehow taken into account here, or could the author suggest how to include this in the framework?

[Figure]

5. I am also interested in the different outcomes of this analysis for precipitation between the daily and the annual time scales (table 4). Maybe one key to interpret this result is that summer vs. winter precipitation patterns are different, and the in-situ network might be more representative of the former than of the latter (or vice versa). I am thinking to convective precipitation here, which sometimes show significantly different elevational gradient from stratiform or orographic precipitation. Some more discussion on precipitation regimes could be interesting in this paper.

SPECIFIC COMMENTS

- Line 149: is this because no gap was originally present in the dataset, or because these gaps were filled? If the second, maybe briefly mention how.

- Section 2.1: a histogram with the elevation distribution of in-situ stations may be helpful, along with more details about the climatology of the study period (annual mean temperature and precipitation, annual runoff etc). Doing so may help the author to set the context of the analysis, especially for non-local readers.

- Section 2.2: is any of these catchments glacierized? If so, how were glaciers considered in this framework? If not, may glaciers hamper the applicability of this method in other regions, especially with regard to the mass-balance-closure term in Eq. 12 and Fig. 6?

- Section 2.3: the approach by Gascoin et al. 2015 was, to my knowledge, developed in the Pyrenees, a mountain range with significantly lower elevations than the Alps. How was the method adapted for the French Alps? Is the performance similar to that originally published by Gascoin et al. 2015 in a different mountain range?

- Line 263ff: was mean precipitation computed across the whole study region? Might doing so exclude more localized precipitation events in favor of more widespread stratiform events?

- Title of Section 4: ASSESSSMENT -> ASSESSMENT

[Figure]

- Section 4: a table with the list of all parameters considered by the snow and hydrologic models would be helpful, including an explicit statement of which parameters where calibrated. Some of these parameters are only mentioned at the very end of the manuscript (Section 5.3).

- Line 321: this should in fact be evaporation to me, since there is no transpiration in the snow module (correct?)

- Section 4.3.1: where the two periods similar in terms of snow conditions and streamflow, as well as mean temperature and mean precipitation across the in-situ network?

- Eq 12 and Section 4.3.2: does the third component of the OF assume that interannual variability in subsurface storage is negligible? This might not be an issue in the studied area, but it may be worth mentioning this in case interested readers would like to apply this approach somewhere else. In fact, results in Section 5.3 do suggest that interannual sub-surface dynamics are worth discussing.

- Section 5.1: do statistics reported in Fig. 4 and at lines 406ff consider areas outside the studied catchments too, including Italy and Switzerland? It might be better to report statistics for the French Alps only here since this is where data were available to this study.

- Section 5.2: the first paragraph of this section and Table 5 should be moved to the Methods. It should also be clarified that each re-calibration mode included hydrologic parameters too (correct?)

- Fig. 6: it seems like all data are within the boundaries given by the water and energy limit. I am not an expert of this approach and was wondering why one should aim to obtain "the least stretched and dispersed cluster". More details on this might be helpful for other readers too.

- Line 490ff and other similar passages of the manuscript: in fact, this result suggests to me that correcting for precipitation and temperature distribution has a stronger impact

on model predictions than adjusting for other snow-related processes like phase partitioning or melt, rather than that "adapting to local snow processes is not indispensable". To me, other processes are important too, but correctly estimating total accumulation is likely the most important one here.

- Section 5.3: I would probably add more details about how parameter identifiability is quantified from Figure 8.

- Line 610 and, earlier, line 490: how were these "physical or general values" obtained?
* * *

---

## Author Comment (AC1) · 3 Feb 2020

**Responses to comments from anonymous Referee 1**

On "Should altitudinal gradients of temperature and precipitation inputs be inferred from key parameters in snow-hydrological models?" by D. Ruelland (HESS-2019-556)

**Referee's comment**

*This paper presents a nicely conceived study where several alternative approaches for distributing temperature and precipitation are compared in a mountain region (French Alps). In addition to standard interpolation approaches based on inverse-distance weighting and Kriging, the author explores the possibility of optimizing lapse rates as part of a snow-hydrologic-model calibration procedure. Results of a split-sample test show that the latter approach provides improved results for the target variables considered during calibration, that is, fractional snow cover (FSC) from MODIS, streamflow, and the water balance. Also, optimizing the temperature and precipitation distribution algorithm with hydrologic data results in the temperature and precipitation fields being colder and wetter than those obtained by using only in-situ measurements of temperature and precipitation, respectively; this result agrees with expectations, especially since the considered ground-based network is not representative of high elevations in the study catchments.*

*I enjoyed reading the manuscript and I think it represents an interesting contribution for HESS, especially because the investigated topic is a clear open issue in mountain hydrology. I do have several general and specific comments, which I attach below. Overall, I think that the revision is feasible.*

**Authors' response**

I would like to sincerely thank the referee for the time and effort he/she spent in reading the initial manuscript and for making many clear, pertinent and constructive suggestions for improvement. This helped a lot to re-write the paper.

**General comments**

**Referee's comment**

*1. in my understanding, the author essentially compares two main strategies: the first is to regionalize temperature and precipitation based on in-situ data and various interpolation/extrapolation schemes based on IDW or Kriging (Section 3); the second is to "embed" part of the distribution process into the snow-hydrologic models via lapse rates that will correct a first-guess distribution based on IDW (see Section 4, Table 2 and 5 and Equations 9 and 10). While the first approach is independent from hydrologic data like fractional snow cover and streamflow, the second does take advantage of these data to adjust some of the distribution parameters. In my understanding, the main point of the paper is that the second strategy is superior to the first, especially since adjusting snow parameters rather than precipitation-distribution parameters does not allow the model to significantly improve its performance (see Table 5 and Fig. 5). Unless I am missing something here, this improvement was however assessed based on the same hydrologic variables that were used to calibrate the snow-hydrologic models, rather than on independent measurements of the two variables of interest: temperature and precipitation. This left me wondering if this experiment shows that "calibrating the local gradients using an inverse snow-hydrological modelling framework" improves actual temperature and precipitation estimates, or if it shows that it improves hydrologic predictions. In principle, one would expect the obtained altitudinal gradients to be both more effective in terms of hydrologic predictions and in terms of temperature and precipitation, but the improvement obtained by "embedding" part of the distribution process into the snow-hydrologic models is quantified in terms of modeling skills for fractional snow cover, streamflow, and the water balance (Figs. 5 to 8) rather than for independent estimates of temperature and precipitation. If independent data of temperature and precipitation at high elevations are not available, then I would recommend the*

*author to clarify the extent to which these results apply to temperature and precipitation in addition to hydrologic variables.*

**Authors' response and modifications to manuscript**
I agree that an improved fit for hydrologic variables may not automatically mean that the model is also better representing weather patterns of temperature and precipitation. Since independent data of temperature and precipitation at high elevations were indeed not available, it was not possible to clarify the extent to which these results apply to temperature and precipitation in addition to hydrologic variables. As a result and following the relevant referee comment and argumentation, the following text has been added in the Section 6.2. Recommendations:

"....However the differences in the two compared approaches are worth discussing. The first is to regionalize temperature and precipitation based on in-situ data and various interpolation/extrapolation schemes based on IDW or Kriging; the second is to "embed" part of the distribution process into the snow-hydrologic models via calibrated lapse rates correcting a first-guess distribution based on IDW. While the first approach is independent from hydrological data like fractional snow cover and streamflow, the second does take advantage of these data to adjust some of the distribution parameters. The second strategy prove superior to the first, especially since calibrating distribution parameters rather than adjusting snow parameters allowed the models to significantly improve their performance. This improvement was however assessed based on the same hydrological variables that were used to calibrate the snow-hydrologic models, rather than on independent measurements of temperature and precipitation. This left wondering if improving hydrologic predictions by calibrating the local gradients using an inverse snow-hydrological modelling framework also improves actual temperature and precipitation estimates. In principle, one would expect the obtained altitudinal gradients to be both more effective in terms of hydrologic predictions and in terms of temperature and precipitation, but the improvement obtained by "embedding" part of the distribution process into the snow-hydrologic models was only quantified in terms of modelling skills. An improved fit for hydrologic variables may not automatically mean that the model is also better representing weather patterns of temperature and precipitation. A good example is that the optimized lapse rates (Fig. 10) can locally be quite different between the two hydrologic models considered. Since independent data of temperature and precipitation at high elevations are not available, we were not able to clarify the extent to which these results apply to temperature and precipitation in addition to hydrologic variables."

**Referee's comment**
*2. The point above is particularly important since hydrologic models may suffer from several sources of conceptual and parametric uncertainties, some of which are visible in the interesting Figure 8. It follows that an improved fit for hydrologic variables may not automatically mean that the model is also better representing weather patterns of temperature and precipitation. A good example here is that the obtained lapse rates (Fig. 9) can locally be quite different between the two hydrologic models considered. To me, this may challenge the idea that this approach could be used to "infer local altitudinal gradients from a sparse network of gauges based on key parameters in the snow-hydrological models" (L 592ff). It does suggest that the method improves hydrologic predictions, but implications for actual temperature and precipitation are more elusive to me and should be discussed more extensively.*

**Authors' response and modifications to manuscript**
Agreed. The sentence (L 592 ff) has been removed and an entire paragraph (based on the argumentation from the referee in his/her general comment #1 and #2 has been added in the Section 6.2. Recommendations (see answer to the preceding comment).

- We removed the sentence " we thus suggest using the proposed modelling framework to infer local altitudinal gradients from a sparse network of gauges based on key parameters in the snow-hydrological models"

**Referee's comment**

*3. Related to this, both hydrologic models were used in lumped mode (L340), even if several other modeling approaches explicitly account for spatial variability in hydrologic processes (e.g., raster-based models). At least some discussion on this point would be interesting.*

**Authors' response and modifications to manuscript**

Agreed. This is part of an on-going research. To address the referee comment, the following text has been added at the end of the conclusion section (6.3. Prospects):

"…Finally, it is worth mentioning that spatial variability was only considered along five elevation bands in each catchment since preliminary tests showed no improvement in the hydrologic predictions when applying the SAR in a full distribution mode. However, the SAR was not designed to explicitly account for topographic effects (slope, aspect and shading) on snow redistribution, accumulation and melt (see e.g. Frey and Holzmann, 2015). A grid-based temperature-index model could thus be implemented to include potential clear-sky direct solar radiation at the surface, thus considering both the seasonal variations of melt rates and the geometric effects on melt attributable to terrain (see e.g. Hock, 1999). It would thus be interesting to assess whether accounting for the influence of such effects can further improve the daily hydrologic predictions at the basin scale."

**Referee's comment**

*4. Spatial variability was considered in the snow model, which was implemented along five elevation bands in each catchment. This model does include all fundamental snow processes, but in my understanding does not include a specific provision for wind drift. Relying on FSC from MODIS may sometimes lead to confounding effects in this regard, where wind-driven accumulation and erosion is mistakenly assumed as due to precipitation or melt. Was this somehow taken into account here, or could the author suggest how to include this in the framework?*

**Authors' response and modifications to manuscript**

The snow accounting routine (SAR) is only based on (semi-)distributed temperature and precipitation inputs to simulate the main snow processes related to accumulation and melt. As a result, it does not include a specific provision for wind drift within each catchment, which would probably require an additional distributed input regarding wind speed and direction. Given the challenge to distribute temperature and precipitation from a sparse gauge network, distributing wind (from even more scarce measures than for temperature/precipitation) may be unrealistic at the spatio-temporal scales considered (daily analysis in the French Alps over the period 1998–2016). It would also probably require applying the SAR in a full distribution mode. But, even doing so, it is unlikely that accounting for wind-driven accumulation and erosion would have a significant impact on the streamflow and FSC simulations at the basin scale, because it can be assumed that these processes are somewhat averaged at the basin scale. Although these aspects are worth discussing, it seems difficult to integrate them into the text without adding weight and making the discussion too large and complex. However, in link with the preceding referee comment, note that the following text (dealing also with spatial variability) has been added at the end of the conclusion section:

"…Finally, it is worth mentioning that spatial variability was only considered along five elevation bands in each catchment since preliminary tests showed no improvement in the hydrologic predictions when applying the SAR in a full distribution mode. However, the SAR was not designed to explicitly account for topographic effects (slope, aspect and shading) on snow redistribution, accumulation and melt (see e.g. Frey and Holzmann, 2015). A grid-based temperature-index model could thus be implemented to include potential clear-sky direct solar radiation at the surface, thus

considering both the seasonal variations of melt rates and the geometric effects on melt attributable to terrain (see e.g. Hock, 1999). It would thus be interesting to assess whether accounting for the influence of such effects can further improve the daily hydrologic predictions at the basin scale."

**Referee's comment**

*5. I am also interested in the different outcomes of this analysis for precipitation between the daily and the annual time scales (table 4). Maybe one key to interpret this result is that summer vs. winter precipitation patterns are different, and the in-situ network might be more representative of the former than of the latter (or vice versa). I am thinking to convective precipitation here, which sometimes show significantly different elevational gradient from stratiform or orographic precipitation. Some more discussion on precipitation regimes could be interesting in this paper.*

**Authors' response and modifications to manuscript**

I sincerely did not see how the analysis of seasonal patterns of precipitation (see Figure 1 below) can explain why interpolation performance is improved by the external drift at the annual (and also monthly) time scale, while it is not at the daily time scale. According to me, this only shows that the correlation between precipitation and topography increases with the increasing time aggregation as already reported in other studies (e.g., Bárdossy and Pegram, 2013; Berndt and Haberlandt, 2018). The elevation-dependency of precipitation thus depends significantly on the accumulation time. For instance, if precipitation events do not occur exactly the same day within the surrounding neighboring gauges, the correlation between precipitation and topography may be weak at the daily time scale, whereas it may be more significant at the monthly, seasonal or annual time scale…

The two preceding sentences were added in the Section 5.1 to try to better explain the different outcomes for precipitation between the daily and the annual time scales (Table 4).

[Figure]

**Fig. 1** Seasonal patterns of precipitation with (a) IDW applied to in-situ network (without elevation depend) and (b) IED applied to in-situ network (with elevation dependency via external drift). Values are in mm per season (i.e. 3 months).

**Specific comments**

**Referee's comments in grey – Authors' responses and modifications to manuscript in blue**

*Line 149: is this because no gap was originally present in the dataset, or because these gaps were filled? If the second, maybe briefly mention how.*

These meteorological gauges were selected because no gap was originally present in their original time series from the 1st September 2000 to the 31st of August 2016. The text has been modified to make the purpose clearer.

*Section 2.1: a histogram with the elevation distribution of in-situ stations may be helpful, along with more details about the climatology of the study period (annual mean temperature and precipitation, annual runoff etc). Doing so may help the author to set the context of the analysis, especially for non-local readers.*

Done. Figure 1 was modified to incorporate elevation distributions of in-situ stations, DEM and basins (see below). Details about the "estimated" climatology of the study period are now provided in Table 1 (see below), which has been modified to include mean annual temperature (T), total precipitation (P), snowfall fraction (S) and streamflow (Q) for each basin. Note however that these values are very delicate to provide since they necessarily rely on approximations depending on the method used to distribute temperature and precipitation (in link with the paper issue). This is why they were not included in the initial submitted paper. As indicated in the modified caption of Table 1, catchment areal temperature, total precipitation and snowfall fraction were estimated after calibrating local altitudinal gradients over 2000–2016 using the snow-hydrological inverse approach proposed in the current paper (see Test #4 in Table 5).

[Figure]

**Fig. 1** Study area and data: (a) Location of the selected precipitation, temperature and streamflow stations, as well as elevations from a SRTM digital elevation model (DEM resampled to a grid with 0.5x 0.5 km cells) in the French Alps; (b) Elevation distributions of in-situ stations, DEM and basins.

**Table 1** Streamflow gauging stations and main catchment characteristics. Percentages of glacierized area were estimated from the World Glacier Inventory (NSIDC, 2012). Mean annual precipitation (P), snowfall fraction (S) and temperature (T) were estimated after calibrating local altitudinal gradients over 2000–2016 using the snow-hydrological inverse approach proposed in the current paper (see Test #4 in Table 5).

| Station | River | Area | Glacierized area | Elevations (m.a.s.l.) | | Mean annual precip. (P) | Snowfall fraction (S) | Mean annual temp. (T) | Mean annual streamflow (Q) |
|---|---|---|---|---|---|---|---|---|---|
| | | (km²) | (%) | Min | Max | (mm/yr) | (%) | ( °C) | (mm/yr) |
| Barcelonnette | Ubaye | 549 | 0 | 1132 | 3308 | 802 | 48 | 1.9 | 521 |
| Lauzet-Ubaye | Ubaye | 946 | 0 | 790 | 3308 | 947 | 44 | 3.0 | 654 |
| Beynes | Asse | 375 | 0 | 605 | 2273 | 920 | 16 | 8.7 | 344 |
| Saint-André-Les-Alpes | Issole | 137 | 0 | 931 | 2392 | 965 | 24 | 6.8 | 481 |
| Villar-Lourbière | Séveraisse | 133 | 4 | 1023 | 3623 | 1561 | 47 | 2.3 | 1317 |
| Val-des-Prés | Durance | 207 | 0 | 1360 | 3059 | 836 | 54 | 0.9 | 688 |
| Briançon | Durance | 548 | 1 | 1187 | 3572 | 844 | 51 | 1.7 | 714 |
| Argentière-la-Bessée | Durance | 984 | 3 | 950 | 4017 | 1014 | 52 | 2.1 | 765 |
| Embrun | Durance | 2170 | 2 | 787 | 4017 | 990 | 48 | 2.9 | 693 |
| Espinasses | Durance | 3580 | 1 | 652 | 4017 | 964 | 45 | 3.4 | 654 |
| Villeneuve-d'Entraunes | Var | 132 | 0 | 926 | 2862 | 989 | 37 | 4.8 | 650 |
| Val-d'Isère | Isère | 46 | 9 | 1831 | 3538 | 1245 | 63 | -1.5 | 1119 |
| Bessans | Avérole | 45 | 12 | 1950 | 3670 | 1399 | 66 | -2.4 | 1311 |
| Taninges | Giffre | 325 | 0 | 615 | 3044 | 2031 | 36 | 4.7 | 1771 |
| Vacheresse | Dranse d'Abondance | 175 | 0 | 720 | 2405 | 1669 | 29 | 4.9 | 1088 |
| La Baume | Dranse de Morzine | 170 | 0 | 690 | 2434 | 1636 | 32 | 4.7 | 1285 |
| Dingy-Saint-Clair | Fier | 223 | 0 | 514 | 2545 | 1649 | 26 | 6.5 | 1243 |
| Allèves | Chéran | 249 | 0 | 575 | 2157 | 1486 | 23 | 6.9 | 819 |
| Mizoën | Romanche | 220 | 9 | 1057 | 3846 | 1205 | 56 | 0.8 | 978 |
| Allemond | L'Eau Dolle | 172 | 2 | 713 | 3430 | 1460 | 46 | 2.7 | 1164 |

*Section 2.2: is any of these catchments glacierized? If so, how were glaciers considered in this framework? If not, may glaciers hamper the applicability of this method in other regions, especially with regard to the mass-balance-closure term in Eq. 12 and Fig. 6?*

During the catchment selection process, we tried to minimize possible interactions with non-snow related processes that could also influence streamflow. Therefore, we tried avoiding glacierized basins, basins with known inter-catchment groundwater flows, and catchments with documented flow diversions. However, it must be acknowledged that some basins are partly glaciated. Table 1 now includes the percentage of glaciated areas estimated for each catchment from the World Glacier Inventory (NSIDC, 2012).

As the majority of basins had negligible glacierized areas (see Table 1), no specific glacier model was activated. This led to ignore the late summer contribution of glacier melt to river discharge in the three basins having 9–12% glacierized areas, which did not affect significantly the mass-balance-closure term at the annual scale. However, it is worth mentioning that more important contribution of ice melt would require a glacier component to not hamper the use of WB in the objective function.

The last paragraph was inserted at the end of the 4.3.2 Section to highlight on the fact that, without activating a glacier model, the applicability of the mass-balance-closure term may be hampered on catchments with an important contribution of ice melt.

*Section 2.3: the approach by Gascoin et al. 2015 was, to my knowledge, developed in the Pyrenees, a mountain range with significantly lower elevations than the Alps. How was the method adapted for the French Alps? Is the performance similar to that originally published by Gascoin et al. 2015 in a different mountain range?*

The referee is right. The approach by Gascoin et al. (2015), itself inspired by previous works from Parajka and Blöschl (2008) in Austria, and Gafurov and Bárdossy (2009) in Afghanistan, was adapted for the French Alps. To do so, the MODIS snow-products were gap-filled in various mountain ranges (including the Pyrenees and the French Alps). The missing data (due mainly to cloud obscuration) were less important in the Pyrenees (46%) than in the Alps (52%) for the same period (2000–2016). As a result, for the temporal deduction by sliding the time filter, we allowed the window size to be incremented up to 9 days in the Alps (versus up to 6 days in the Pyrenees) to account for the differences in cloud obscuration. Moreover, Gascoin et al. (2015) used an adjacent spatial deduction as a second step: each no-data pixel was reclassified as snow (no-snow) if at least five of the eight adjacent pixels were classified as snow (no-snow). We did not use this step considering it was a too rough approximation. Finally, we validated our adapted method on the two mountain ranges. Validation based on confusion matrices with 1 image/month (i.e. about 200 cloud-free images over the studied period) showed that the gap-filling technique applied to the MODIS snow-products led to the reconstruction of validation images with average accuracies of 98% in the Pyrenees and 94% in the Alps.

To address the referee comment, more details regarding the gap-filling method and validation have been given through additional text and a new figure (see below) in the Section 2.3.

[revised manuscript text omitted]

*Line 263ff: was mean precipitation computed across the whole study region? Might doing so exclude more localized precipitation events in favor of more widespread stratiform events?*

In the initial submission, parameters were only estimated for days with at least 1 mm mean precipitation, i.e. approximatively 41% of the daily sample (whereas parameters were calculated for all months and years since there were no locations with dry months or dry years in the dataset). The 1 mm mean precipitation was indeed computed across the whole study region. The initial motivation was to limit the effect of precipitation zero values during the optimization of the daily local regression between precipitation and elevation for external drift. Doing so might indeed bias the optimization of the $n(u)$ surrounding observations during the leave-one-out cross-validation, thus potentially affecting the computation of the external drift with the KED and IED technique when

further used to interpolate daily precipitation in the study region (Fig. 5 – Fig. 4 in the initial submission).

Following the referee comment, the leave-one-out cross-validation was re-run with the whole set of daily samples using the four interpolation techniques (IDW, ORK, KED and IED). As a result, the RMSE, MAE and NSE values changed for the cross-validation of the interpolation methods against precipitation daily series (see new values in Table 4). However, interestingly, this did affect neither the ranking between methods nor the optimized interpolation parameters. For instance, 17 surrounding neighbors were still found during optimization to compute altitudinal gradients of precipitation based on the daily, local linear regression with KED and IED. This means that the initial results (optimized parameters and daily interpolated precipitation) were not affected by the 41% subset. In other words, the subset did not exclude more localized precipitation events in favor of more widespread events. However, as it revealed unnecessary, the sentence about it in the section 3.3 was deleted and the RMSE, MAE and NSE scores were changed in Table 4 accordingly to the computation of the cross-validation with the all daily precipitation samples.

*Title of Section 4: ASSESSSMENT -> ASSESSMENT.*
Corrected in the revised manuscript.

*Section 4: a table with the list of all parameters considered by the snow and hydrologic models would be helpful, including an explicit statement of which parameters where calibrated. Some of these parameters are only mentioned at the very end of the manuscript (Section 5.3).*
The referee probably missed Table2 that lists all (fixed or calibrated) parameters regarding the snow accounting routine (SAR) and Table3 that lists all calibrated parameters regarding the two hydrological models. The two models were run with the SAR on top. Since the SAR parameters were calibrated differently (from 2 to 5 free parameters) in each modelling experiment, it was necessary to introduce two separated tables for the parameters. Note however that these two tables are presented in two consecutive sections, making them theoretically understandable.

*Line 321: this should in fact be evaporation to me, since there is no transpiration in the snow module (correct?)*
There is no transpiration in the snow module. However, PE is computed based on the temperature-based formulation proposed by Oudin et al. (2005) which explicitly mentioned the "evapotranspiration" term ( "…towards a simple and efficient potential evapotranspiration model for rainfall-runoff modelling"). The formula aims at estimating the maximum amount of evapotranspirable water taking into account the meteorological context and for a plant cover corresponding to grass. As a result, it is common to use the term potential evapotranspiration when one refers to the Oudin formula.

*Section 4.3.1: where the two periods similar in terms of snow conditions and streamflow, as well as mean temperature and mean precipitation across the in-situ network?*
Mean annual precipitation increased by around 17% between the two periods (1104 mm/yr vs. 1294 mmm/yr), while mean annual temperature were stable (9.2 °C vs. 9.3 °C) across the in-situ network presented in Section 2.1. Although the second period was generally wetter, this hides differences in between the catchments. At the basin scale, the differences between the two periods ranged from -10% to 15% for precipitation, -0.5 °C to +0.5 °C for temperature, and -11% to +50% for streamflow. These details are now indicated in Section 4.3.1.

As far as the snow conditions are concerned, differences in mean annual snowfall between the two periods ranged from -10% to +50% according to the best-performing simulations. As these values were obtained through simulations, we feel it is not correct to mention them in Section 4.3.1. We hope the above values regarding precipitation, temperature and streamflow between the two periods are sufficient to address the referee comment.

*Eq 12 and Section 4.3.2: does the third component of the OF assume that interannual variability in subsurface storage is negligible? This might not be an issue in the studied area, but it may be worth mentioning this in case interested readers would like to apply this approach somewhere else. In fact, results in Section 5.3 do suggest that interannual sub-surface dynamics are worth discussing.*

The referee is right. The third component (WB) of the objective function assumes that interannual variability in subsurface storage is negligible. This has been indicated in the text as required in case readers would like to apply the approach somewhere else. To address another referee comment, it was also highlighted that, without activating a glacier model, the applicability of the mass-balance-closure term may be hampered on catchments with an important contribution of ice melt.

Section 5.3 does not suggest that "interannual" sub-surface dynamics are worth discussing. Rather, it suggests that groundwater exchanges may occur by gaining/loosing water from/to neighboring basins during the year. GR4J can account for part of these exchanges through its X2 parameter whereas HBV9 considers catchments as closed systems. Despite these differences, snow-hydrological predictions are significantly improved for both HBV9 and GR4J. To address the referee comment, the following sentence was slightly rephrased in the 5.4 section:

"…The differences between the two models may be due to the GR4J ability to gain (or loose) water from inter-catchment groundwater flows through its X2 parameter (see section 5.3.), unlike HBV9 which considers the catchment as a closed system. On the other hand, HBV9 relies on more parameters for production and transfer, thus enabling to compensate differently for the errors in the precipitation volumes…"

*Section 5.1: do statistics reported in Fig. 4 and at lines 406ff consider areas outside the studied catchments too, including Italy and Switzerland? It might be better to report statistics for the French Alps only here since this is where data were available to this study.*

Yes. To address the referee comment, a geographical mask (representing only the six French administrative departments from which data were selected) was applied on the initial maps. Statistics were re-computed accordingly. Note that only means have changed since the minimum and maximum values were still in the masked maps. Note that the same mask was also applied to the DEM in Fig. 1 to make coherent the presentation.

*Section 5.2: the first paragraph of this section and Table 5 should be moved to the Methods. It should also be clarified that each re-calibration mode included hydrologic parameters too (correct?)*

We acknowledge that this first paragraph and Table 5 could somehow be moved to the Methods. However, as indicated in the text, "for sake of brevity, here we only present the results we obtained with the datasets interpolated with the IDW and IED procedures, since cross-validation at the daily time scale showed that they slightly outperformed the ORK and KED methods, respectively". This means that the short "methodological" paragraph and associated Table 5 are strongly linked to the results presented in the previous section 5.1 and cannot be moved to the Methods since they depend on these results. Note also that all the modelling experiment (snow-accounting routine, hydrological models and calibration/validation methods) are fully described in the Methods. Here, we further combine this modelling experiment to part of the results of the interpolation methods to introduce the tests to account for elevation dependency in the T and P inputs via the modelling experiment. Since the tests (#1-6) are rather complex and since their results are immediately presented and discussed in the Section 5.2, we believe that reading is probably easier in that way.

We hope these arguments will convince the referee.

Regarding the second comment, the parameters of the SAR and the hydrological model were indeed optimized simultaneously, as already indicated in Section 4.3.2. Following the referee comment, it was however repeated and clarified that each re-calibration mode included hydrologic parameters too, by adding the following text in the Table 5 caption: "Note that each calibration tests included also the hydrological parameters of GR4J or HBV9 (the parameter ranges tested are listed in Table 2 for the SAR and in Table 3 for the hydrological models)."

*Fig. 6: it seems like all data are within the boundaries given by the water and energy limit. I am not an expert of this approach and was wondering why one should aim to obtain "the least stretched and dispersed cluster". More details on this might be helpful for other readers too.*

A closer look to the Figure shows that the dots are systematically within the boundaries given by the water and energy limits only with Test #4. It can also be observed that Test #4 led to the least stretched and dispersed cluster, which was not particularly intended. This is only a finding which adds to the more important fact that the dots are all located within the "physical" limits by the water and energy limits. As this comment seemed to cause confusion, it has been removed. The text has also been modified as follows to try to better explain the differences in water balance between the tests:

"…Water balance is ensured for each year in each catchment only with Test #4. Indeed, unlike the other simulations (Tests #1, #2 and #3), all the dots are within the boundaries given by an upper water limit where $Q = P$ (i.e., $y = Q/P = 1$) and a lower energy limit where $Q = P − PE$ ($Q/P = 1 − PE/P$ $\leftrightarrow y = 1 − 1/x$). This means that annual simulated runoff never exceeds total precipitation and that annual runoff deficit never exceeds total $PE$. Altitudinal temperature and precipitation gradients inferred from snow-hydrological modelling thus lead to more realistic catchment water balance than when they are estimated from gauges using interpolation."

*Line 490ff and other similar passages of the manuscript: in fact, this result suggests to me that correcting for precipitation and temperature distribution has a stronger impact on model predictions than adjusting for other snow-related processes like phase partitioning or melt, rather than that "adapting to local snow processes is not indispensable". To me, other processes are important too, but correctly estimating total accumulation is likely the most important one here.*

Done. The sentence has been modified as follows (see also other modifications in link with other comments notably in Section 6.2 Recommendations):

'…This suggests that correcting for temperature and precipitation distribution has a stronger impact on model predictions than adjusting for snow-related processes like phase partitioning or melt, and that correctly estimating total accumulation is likely to play a first-order role in the snow-hydrological responses of the studied catchments.'

*Section 5.3: I would probably add more details about how parameter identifiability is quantified from Figure 8.*

Agreed. Figure 8 (now Figure 9) was slightly modified to indicate more clearly which parameters belong to the SAR or to the hydrological models (GR4J). Moreover, more details have been provided in the figure caption and in the text. Variation coefficients (in %) of the 20% best parameter solutions compared to the optimised values for each parameter have also been introduced in the Figure and in the text to bring more deails about how parameter identifiability can be "quantified" from the Figure.

*Line 610 and, earlier, line 490: how were these "physical or general values" obtained?*

To make the purpose clearer, the initial sentence has been completed in Section 4.1 as follows:

"….The aim of using this mode was to account for elevation dependency in the T and P inputs from constant, calibrated orographic gradients while fixing the parameters that control snow accumulation and melt to physical or general values: precipitation phase determined based on a linear separation between -1 °C and +3 °C (see USACE, 1956), temperature threshold for snowmelt fixed at 0 °C, degree-day melt factor set at 5 mm. $°C^{-1}.d^{-1}$ (mean general value taken from Hock, 2003)."

The additional references have been also inserted in the reference list.

New references

Frey, S. and Holzmann H.: A conceptual, distributed snow redistribution model, Hydrol. Earth Syst. Sci., 19, 4517–4530, https://doi.org/10.5194/hess-19-4517-2015, 2015.

Gafurov, A. and Bárdossy, A.: Cloud removal methodology from MODIS snow cover product, Hydrol. Earth Syst. Sci., 13, 1361–1373, https://doi.org/10.5194/hess-13-1361-2009, 2009.

Hock, R.: A distributed temperature-index ice- and snowmelt model including potential direct solar radiation. J. Glaciology 45, 101–111, https://doi.org/10.3189/S0022143000003087, 1999.

Hock, R.: Temperature index melt modelling in mountain areas, J. Hydrol., 282, 104–115, https://doi.org/10.1016/S0022-1694(03)00257-9, 2003.

NSIDC: World Glacier Inventory, Version 1. Boulder, Colorado USA. NSIDC: National Snow and Ice Data Center. https://doi.org/10.7265/N5/NSIDC-WGI-2012-02, 2012.

USACE: Snow Hydrology: Summary Report of the Snow Investigation. Portland, Oregon, North Pacific Division, Corps of Engineers, U.S. Army, 1956.

---

## Referee Comment (RC2) · Anonymous Referee #2 · 11 Feb 2020

The article analyzes the sensitivity of a snow accounting procedure and hydrological modeling results to the evaluation of temperature and precipitation in space and time in mountainous catchments. The study is based on a set of 20 catchments in the French Alps and two hydrological models. The author evaluates the interplay between the lapse rate, snow routine and hydrological model parameters.

I found this is a clear and interesting paper. I have a few suggestions for improvement detailed below, some of which are quite major and requiring new calculations. I suggest considering the paper for possible publication in HESS after major revision.

Detailed comments

1. I found that the literature review could have been more exhaustive, to better stress the originality of the work compared to existing studies on similar or close topics. Some recent works could be discussed, for example the work by Le Moine et al. (2015) on the link between snow and hydrological sub-models in model parameterization, some studies on using snow data to calibrate hydrological models (Besic et al. 2014, Henn et al. 2016, Riboust et al. 2019), some studies with physical approaches to estimate lapse rates (Rahman et al. 2014, Zhang et al. 2015, Naseer et al. 2019). The review could also be extended on how gauge undercatch factors are estimated. The author should further discuss to which extent the proposed approach is original compared to these past findings.

2. Section 2.1: It would be useful to add a figure showing the distributions of mean precipitation and temperature over the set of gauges, to give an idea of the variability across the study domain.

3. Section 2.2: Reference could be given to the work by Leleu et al. (2014).

4. Table 1: Please explain the meaning of abbreviations in the last column. Is this information useful here?

5. Section 3.3: The author calculates the efficiency criteria on precipitation values. However, the criteria may be strongly influenced by a few large rainfall events, which may not be representative of the average characteristics of precipitations. It may be useful to consider computing the efficiency criteria on transformed precipitation (e.g. root square transformation) to avoid putting too much weight on outlier values. Would this change something in results?

6. L261: The name "RMSE" given to the normalized RMSE is a bit confusing. The author may choose another name, e.g. NRMSE.

7. Section 4.1: Some modifications in this snow module were recently proposed by Riboust et al. (2019), to account for snow-covered area. This should be shortly commented, to better explain how the proposed approach compares to this existing work.

8. Fig.3: Maybe add the meaning of the key variables (at least inputs/output) in the figure caption. If UZL is the threshold for the upper output, maybe the arrow should stop at the level of this output.

9. L376-378: This is a point I did not understand in the proposed methodology. By introducing this criterion WB in the objective function, the author forces the model to close the water balance in the sense of Budyko. This is quite successful when looking at results shown in Fig. 6, since no data lies outside the boundaries of balance closure in the plot. However, I do not understand the physical rationale behind putting this constraint. There are many catchments where the water balance cannot be closed in the Budyko sense for good reasons, mainly because of underground water exchanges. The author artificially constrains the models using WB. I think a more classical bias criterion would be better to consider instead.

10. Table 4: There is a strong drop in the NSE criterion for temperature when going from monthly to daily time steps for IDW and ORK. How this drop can be explained?

11. L472-476: I think this result is the consequence of using WB in the objective function. As mentioned above, this constraint is artificial and potentially counterproductive for the efficiency of the model.

12. L510-516: I find this a bit contradictory with the WB constraint. If the author makes the hypothesis that underground water exchanges between catchments may play a key role, why does the author constrain water balance not to account for such exchanges in the optimization phase?

13. Fig. 8 is interesting. However there are some cases which reveal that the optimum is probably outside the preset parameter range. This is typically the case for Test#1 for parameters X1 to X3. Therefore the ranges should be extended.

Cited references

[Figure]

Besic, N., et al. (2014). "Calibration of a distributed SWE model using MODIS snow cover maps and in situ measurements." Remote Sensing Letters 5(3): 230-239.

Henn, B., et al. (2016). "Combining snow, streamflow, and precipitation gauge observations to infer basin-mean precipitation." Water Resour. Res. 52(11): 8700-8723.

Le Moine, N., et al. (2015). "Hydrologically Aided Interpolation of Daily Precipitation and Temperature Fields in a Mesoscale Alpine Catchment." J. Hydrometeorol. 16(6): 2595-2618.

Leleu, I., et al. (2014). "Re-founding the national information system designed to manage and give access to hydrometric data." La Houille Blanche(1): 25-32.

Naseer, A., et al. (2019). "Distributed Hydrological Modeling Framework for Quantitative and Spatial Bias Correction for Rainfall, Snowfall, and Mixed-Phase Precipitation Using Vertical Profile of Temperature." J. Geophys. Res.-Atmos. 124(9): 4985-5009.

Rahman, K., et al. (2014). "Streamflow response to regional climate model output in the mountainous watershed: a case study from the Swiss Alps." Environmental Earth Sciences 72(11): 4357-4369.

Riboust, P., et al. (2019). "Revisiting a simple degree-day model for integrating satellite data: implementation of SWE-SCA hystereses." Journal of Hydrology and Hydromechanics 67(1): 70-81.

Zhang, F., et al. (2015). "Snow cover and runoff modelling in a high mountain catchment with scarce data: effects of temperature and precipitation parameters." Hydrol. Processes 29(1): 52-65.
* * *

---

## Author Comment (AC2) · 25 Feb 2020

**Responses to comments from anonymous Referee 2**

On "Should altitudinal gradients of temperature and precipitation inputs be inferred from key parameters in snow-hydrological models?" by D. Ruelland (HESS-2019-556)

**Referee's comment**

*The article analyzes the sensitivity of a snow accounting procedure and hydrological modeling results to the evaluation of temperature and precipitation in space and time in mountainous catchments. The study is based on a set of 20 catchments in the French Alps and two hydrological models. The author evaluates the interplay between the lapse rate, snow routine and hydrological model parameters.*

*I found this is a clear and interesting paper. I have a few suggestions for improvement detailed below, some of which are quite major and requiring new calculations. I suggest considering the paper for possible publication in HESS after major revision.*

**Authors' response**

I would like to thank the referee for the time spent in reviewing the initial paper and making interesting suggestions. Most of them were judged useful even though I did not agree with all comments. In any cases, I provided a point-by-point response to the reviewer's comments and tried to bring modifications to the manuscript accordingly.

**Detailed comments**

**Referee's comment**

*1. I found that the literature review could have been more exhaustive, to better stress the originality of the work compared to existing studies on similar or close topics. Some recent works could be discussed, for example the work by Le Moine et al. (2015) on the link between snow and hydrological sub-models in model parameterization, some studies on using snow data to calibrate hydrological models (Besic et al. 2014, Henn et al. 2016, Riboust et al. 2019), some studies with physical approaches to estimate lapse rates (Rahman et al. 2014, Zhang et al. 2015, Naseer et al. 2019). The review could also be extended on how gauge undercatch factors are estimated. The author should further discuss to which extent the proposed approach is original compared to these past findings.*

**Authors' response and modifications to manuscript**

I thank the referee for these additional references, some of which I did not know. Note however that some recent references have not been yet published at the moment where the current paper was conceived and written (e.g. Naseer et al., 2019; Riboust et al., 2019), which makes it difficult to provide an up-to-date literature.

Most of the proposed references were judged useful. Consequently, they were cited in the text and added to the reference list.

Following the referee comment, the following paragraph was added in the introduction section:

"…Several studies proposed approaches to estimate lapse rates based on physically-based or conceptual models on specific catchments. Zhang et al. (2013) showed that the runoff simulation results involving snowmelt and rainfall runoff were highly sensitive to the temperature and precipitation lapse rates in a Tibetan catchment. Rahman et al. (2014) calibrated the SWAT model in a snow-dominated basin in the Swiss Alps and found also that temperature lapse rate was significantly important for hydrological performance. Naseer et al. (2019) considered a dynamic lapse rate based on a vertical profile of temperature in a catchment in Japan and succeeded to improve the precipitation phase in a distributed hydrological modelling framework. Henn et al. (2016) investigated the value of snow data to constrain the inference of precipitation from streamflow, using lumped hydrologic models and an elevation-band snow model in a Californian basin. Their

results suggested that multiple types of hydrologic observations, such as streamflow and SWE, may help to constrain the water balance of high-elevation basins. Le Moine et al. (2015) proposed a calibration strategy where the parameters of both an interpolation model and a daily snow-hydrological model are jointly inferred in a multi-variable approach applied in a catchment in the French Alps. Using a hydro-meteorological modelling chain involving 31 calibrated parameters, they showed the potential of using different types of observations (rain gauges, snow water equivalent measurements and streamflow data) to help assess temperature and precipitation lapse rates according to different weather types. These examples encourage testing whether an inverse modelling approach based on calibrated constant lapse rates can perform well with parsimonious conceptual models applied in numerous basins."

The recent reference from Riboust et al. (2019) was mentioned later in the introduction:
"…Moreover, other authors (Franz and Karsten, 2013; He et al., 2014; Riboust et al., 2019) showed that adding snow data information to the calibration procedure enabled the identification of more robust snow parameter sets by making the snow models less dependent on the rainfall-runoff model with which they are coupled.
It was also acknowledged in section 4.1 about the snow accounting routine (se answer to the referee comment #7.

Some references were also acknowledged in the conclusion section:
"…Accurate estimate of these parameters greatly helps in determining the form of precipitation and spatial distribution of temperature and precipitation, and are critical for snow cover and runoff modelling in high mountain catchments, as already reported in other regions (Zhang et al., 2013; Naseer et al., 2019)."

**Referee's comment**
*2. Section 2.1: It would be useful to add a figure showing the distributions of mean precipitation and temperature over the set of gauges, to give an idea of the variability across the study domain.*

**Authors' response and modifications to manuscript**
The other reviewer also made this comment. Details about the "estimated" climatology of the study period are now provided in Table 1 (see below), which has been modified to include mean annual temperature (T), total precipitation (P), snowfall fraction (S) and streamflow (Q) for each basin. Note however that T, P and S values are very delicate to provide since they necessarily rely on approximations depending on the method used to distribute temperature and precipitation (in link with the paper issue). This is why they were not included in the initial submitted paper. As indicated in the modified caption of Table 1, catchment areal temperature, total precipitation and snowfall fraction were estimated after calibrating local altitudinal gradients over 2000–2016 using the snow-hydrological inverse approach proposed in the current paper (see Test #4 in Table 5).

**Table 1** Streamflow gauging stations and main catchment characteristics. Percentages of glacierized area were estimated from the World Glacier Inventory (NSIDC, 2012). Mean annual precipitation (P), snowfall fraction (S) and temperature (T) were estimated after calibrating local altitudinal gradients over 2000–2016 using the snow-hydrological inverse approach proposed in the current paper (see Test #4 in Table 5).

| Station | River | Area | Glacierized area | Elevations (m.a.s.l.) | | Mean annual precip. (P) | Snowfall fraction (S) | Mean annual temp. (T) | Mean annual streamflow (Q) |
|---|---|---|---|---|---|---|---|---|---|
| | | (km²) | (%) | Min | Max | (mm/yr) | (%) | ( °C) | (mm/yr) |
| Barcelonnette | Ubaye | 549 | 0 | 1132 | 3308 | 802 | 48 | 1.9 | 521 |
| Lauzet-Ubaye | Ubaye | 946 | 0 | 790 | 3308 | 947 | 44 | 3.0 | 654 |
| Beynes | Asse | 375 | 0 | 605 | 2273 | 920 | 16 | 8.7 | 344 |
| Saint-André-Les-Alpes | Issole | 137 | 0 | 931 | 2392 | 965 | 24 | 6.8 | 481 |
| Villar-Lourbière | Séveraisse | 133 | 4 | 1023 | 3623 | 1561 | 47 | 2.3 | 1317 |

| | | | | | | | | | |
|---|---|---|---|---|---|---|---|---|---|
| Val-des-Prés | Durance | 207 | 0 | 1360 | 3059 | 836 | 54 | 0.9 | 688 |
| Briançon | Durance | 548 | 1 | 1187 | 3572 | 844 | 51 | 1.7 | 714 |
| Argentière-la-Bessée | Durance | 984 | 3 | 950 | 4017 | 1014 | 52 | 2.1 | 765 |
| Embrun | Durance | 2170 | 2 | 787 | 4017 | 990 | 48 | 2.9 | 693 |
| Espinasses | Durance | 3580 | 1 | 652 | 4017 | 964 | 45 | 3.4 | 654 |
| Villeneuve-d'Entraunes | Var | 132 | 0 | 926 | 2862 | 989 | 37 | 4.8 | 650 |
| Val-d'Isère | Isère | 46 | 9 | 1831 | 3538 | 1245 | 63 | -1.5 | 1119 |
| Bessans | Avérole | 45 | 12 | 1950 | 3670 | 1399 | 66 | -2.4 | 1311 |
| Taninges | Giffre | 325 | 0 | 615 | 3044 | 2031 | 36 | 4.7 | 1771 |
| Vacheresse | Dranse d'Abondance | 175 | 0 | 720 | 2405 | 1669 | 29 | 4.9 | 1088 |
| La Baume | Dranse de Morzine | 170 | 0 | 690 | 2434 | 1636 | 32 | 4.7 | 1285 |
| Dingy-Saint-Clair | Fier | 223 | 0 | 514 | 2545 | 1649 | 26 | 6.5 | 1243 |
| Allèves | Chéran | 249 | 0 | 575 | 2157 | 1486 | 23 | 6.9 | 819 |
| Mizoën | Romanche | 220 | 9 | 1057 | 3846 | 1205 | 56 | 0.8 | 978 |
| Allemond | L'Eau Dolle | 172 | 2 | 713 | 3430 | 1460 | 46 | 2.7 | 1164 |

**Referee's comment**

*3. Section 2.2: Reference could be given to the work by Leleu et al. (2014).*

**Authors' response and modifications to manuscript**

I could not access this reference from the journal "*La Houille Blanche*", although I contacted the authors to obtain a hard copy. As a result, I could not judge if it was appropriate to reference this work here. However, Section 2.2 deals mainly with the streamflow series gathered from the French hydrological database (www.hydro.eaufrance.fr) for 20 catchments, as indicated in the text. In other publications, the French hydrological database is usually acknowledged this way by indicating the web site from which data (and metadata) can be freely accessed.

**Referee's comment**

*4. Table 1: Please explain the meaning of abbreviations in the last column. Is this information useful here?*

**Authors' response and modifications to manuscript**

The information was indeed not very useful. It has been removed from the revised Table 1 (see Table above).

**Referee's comment**

*5. Section 3.3: The author calculates the efficiency criteria on precipitation values. However, the criteria may be strongly influenced by a few large rainfall events, which may not be representative of the average characteristics of precipitations. It may be useful to consider computing the efficiency criteria on transformed precipitation (e.g. root square transformation) to avoid putting too much weight on outlier values. Would this change something in results?*

**Authors' response and modifications to manuscript**

The Jack-Knife cross-validation procedure on precipitation series is usually performed on RMSE (e.g. Kyriadis et al., 2001; Le Moine et al., 2013; Yang et al., 2018 to cite just a few). I could not find in the literature an example where RMSE was applied based on a root square transformation of precipitation in such a procedure. Of course, it can be argued than computing the objective function (here RMSE) on transformed precipitation may lead to different interpolation parameters. However, it can also be assumed that large rainfall events are critical for elevation/precipitation regressions when looking at the optimized surrounding gauges to consider in the KED and IED methods. This means that using an efficiency criterion on transformed precipitation may also put less weight on large rainfall events to compute the regressions.

Following the referee comment, the JK cross-validation was re-run with IED using RMSE$_{sqrt}$ on daily, monthly and yearly precipitation series. No significant differences could be found in the optimized interpolation parameters as shown in the following Table 1. Only the number of optimized number of surrounding neighbours changed slightly at the daily time scale from 17 to 15. This cannot be not judged significant as this parameter presents a low sensitivity between 12 and 20 neighbours due to the intrinsic compromise looking on the whole study domain.

**Table 1** Cross-validation of the IED method with RMSE or RMSE$_{sqrt}$ (root square transformation of precipitation) against yearly, monthly and daily series from precipitation gauges over the period 2000–2016. The values of $n(u)$ and $\omega$ represent the interpolation parameters, which were optimised using the leave-one-out procedure.

|  | Current paper (RMSE as EC) | | | Alternative test (RMSE$_{sqrt}$ as EC) | | |
|  | Efficiency criterion | IED parameters | | Efficiency criterion | IED parameters | |
|  | RMSE | $n(u)$ | $\omega$ | RMSE$_{sqrt}$ (RMSE) | $n(u)$ | $\omega$ |
| Yearly | 150.31 mm/year | 12 | 3 | 2.13 (150.31) | 12 | 3 |
| Monthly | 22.20 mm/month | 12 | 2 | 1.05 (22.2) | 12 | 2 |
| Daily | 2.90 mm/day | 17 | 2 | 0.51 (2.91) | 15 | 2 |

For these different reasons, and also because the RMSE$_{sqrt}$ is difficult to interpret since units are not allowed, the usual RMSE criterion was kept in the article as efficiency criteria on precipitation series. Note also that the two other criterions (MAE and NSE which were not used for optimization) are not presented anymore in Table 4 as they seemed to cause confusion in the result interpretation (see answer to the Referee's comment #10 below).

**Referee's comment**
*6. L261: The name "RMSE" given to the normalized RMSE is a bit confusing. The author may choose another name, e.g. NRMSE.*

**Authors' response and modifications to manuscript**
There was an error in the text, which is now corrected. In fact the models were cross-validated against the usual RMSE (root mean square error) without any normalization, as follows:

$$\text{RMSE} = \sqrt{\sum_{i=1}^{N}(Vpre_i - Vobs_i)^2/N} \tag{1}$$

where $Vpre_i$ and $Vobs_i$ are the predicted and observed variables respectively at time scale *i* and *N* the total number of time steps.

**Referee's comment**
*7. Section 4.1: Some modifications in this snow module were recently proposed by Riboust et al. (2019), to account for snow-covered area. This should be shortly commented, to better explain how the proposed approach compares to this existing work.*

**Authors' response and modifications to manuscript**
Agreed. This is now commented in the Section 4.1, as follows:

"In the original version of CEMANEIGE, fractional snow-covered area (FSC) is calculated as follows:

$$FSC_i(t) = \min\left(\frac{SWE_i(t)}{SWE_{th}}, 1\right) \tag{11}$$

where *SWE* is the quantity of snow accumulated on the catchment in snow water equivalent (a state variable of the model, in mm), and $SWE_{th}$ is the model's melting threshold. $SWE_{th}$ is calculated as

being equal to 90% of mean annual solid precipitation on the catchment considered (Valéry et al., 2014). Alternative approaches have been proposed to account for the hysteresis that exists between *FSC* and *SWE* during the accumulation and melt phases (Riboust et al., 2019). However, introducing such a hysteresis adds two additional free parameters to the SAR. Instead, $SWE_{th}$ was fixed to 40 mm since preliminary sensitivity analyses showed that this value gave very satisfactory *FSC* values when compared to the MODIS observations in the studied catchments."

**Referee's comment**
*8. Fig.3: Maybe add the meaning of the key variables (at least inputs/output) in the figure caption. If UZL is the threshold for the upper output, maybe the arrow should stop at the level of this output.*

**Authors' response and modifications to manuscript**
Agreed. The meaning of the key variables has been added in the Figure caption, as follows: "*R*, *M*, *PE* and *Q* stand for rainfall, melt, potential evapotranspiration and streamflow, respectively". Following the referee comment, the arrow for *UZL* was also slightly modified in the figure.

**Referee's comment**
*9. L376-378: This is a point I did not understand in the proposed methodology. By introducing this criterion WB in the objective function, the author forces the model to close the water balance in the sense of Budyko. This is quite successful when looking at results shown in Fig. 6, since no data lies outside the boundaries of balance closure in the plot. However, I do not understand the physical rationale behind putting this constraint. There are many catchments where the water balance cannot be closed in the Budyko sense for good reasons, mainly because of underground water exchanges. The author artificially constrains the models using WB. I think a more classical bias criterion would be better to consider instead.*

**Authors' response and modifications to manuscript**
The rationale behind using WB in the objective function was to enhance the parameter identifiability without decreasing the model efficiency. I agree that there are many catchments where the water balance cannot be closed in the Budyko sense for good reasons due notably to inter-catchment groundwater exchanges (IGE). There are also several bad reasons for which water balance is not closed at the basin scale: errors in the precipitation volumes, wrong estimate of potential evapotranspiration, inaccurate knowledge of the catchment area, etc. Since the paper deals with the lapse rates of the temperature and precipitation inputs, it can be assumed that using a more classical objective function (i.e. without WB) may also lead to optimize the lapse rates while water balance is not closed for the above "bad" reasons (errors in the precipitation volumes, wrong estimate of potential evapotranspiration). In the paper, two models are used: HBV which considers the catchments as closed systems, and GR4J which allows for potential IGE (and/or wrong water balance estimates) to be considered via its X2 parameter. Consequently, the Budyko constraint left room for not completely close the water balance with GR4J.

Nevertheless, I decided to follow the referee comment because sensitivity analyses to the objective function are far beyond the paper issue and because other readers may not be convinced by the proposed constraint without such an in-deep demonstration. I re-run all the simulations with a more classical objective function (i.e. without using the WB constraint) based only on $NSE_{FSC}$ and $NSE_{sqrtQ}$. The results were the same as regards to the modelling distribution performances between the various tests (see Figures 1 and 2 below). Changing the objective function thus did not change the main findings of the paper notably as regards to the interest of calibrating the temperature and precipitation lapse rates via a parsimonious 2-parameter SAR. Obviously, the water balance was not closed systematically and it was not interesting anymore to present the Budyko graphs for the different tests. This led also to deteriorate the general parameter identifiability (see Figure 3 below). This particularly affected the identifiability of the *X2* parameter for the reasons explained above.

However, the ranking of parameter identifiability in between tests did not change and the parsimonious 2-parameter SAR still led to the best parameter identifiability, while remaining among the best-performing models. Finally, the optimized lapse rates were slightly changed: the precipitation gradients were notably found more similar between the two hydrological models tested (see Figure 4 below). This is probably the most important reason that convinced me to renounce to the WB constraint in the objective function (even though I still believe that this constraint is both original and efficient).

**With WB in the objective function**

**Without WB in the objective function**

[Figure]

**Fig. 1** Boxplots (showing 0.00, 0.25, 0.50, 0.75 and 1.00 percentiles) of the efficiency distributions obtained in validation by the (a) GR4J and (b) HBV9 models combined with the snow model according to six different tests (see Table 5) to account for elevation dependency in the *T* and *P* inputs on the 20 snow-affected Alpine catchments.

**With WB in the objective function**

[Figure]

[Figure]

**Fig. 2** Comparison of snow-hydrological simulations with elevation dependency according to Tests #1 to #4 (see Table 5) with GR4J for the Durance at Serre-Ponçon. The graphs show mean inter-annual time-series of temperature, precipitation, streamflow and fractional snow cover at the catchment scale in validation over the period 2008–2016. $T_{mean}$, $P_{mean}$ and $S_{mean}$ stand for mean annual temperature, precipitation, and snowfall, respectively. The efficiency criterions $NSE_{SNOW}$, $NSE_Q$, $NSE_{lnQ}$ and $VE_C$ are computed from continuous (not mean seasonal) series over 2008–2016.

[Figure]

[Figure]

**Fig. 3** Parameter sensitivity to the objective function (OF) according to Tests #1 to #6 (see Table 5) with GR4J combined with the snow accounting routine (SAR) on the Durance at Serre-Ponçon. The values and dots in red indicate the optimised calibrated parameters when minimising OF, the black dots represent trials of the SCE-UA optimisation algorithm, and the values in blue are the variation coefficients (in %) of the 20% best parameter solutions compared to the optimised values for each parameter (the lowest value, the easiest parameter identifiability). Note that depending on the tests, the calibrated parameters of the SAR vary from 2 to 5 (see Table 5 and Table 2 in the manuscript), while the GR4J hydrological models has 4 parameters (see Table 3 in the manuscript).

[Figure]

**Fig. 4** Boxplots (showing 0.00, 0.25, 0.50, 0.75 and 1.00 percentiles) of the ranges of (a) temperature and (b) precipitation lapse rates calibrated with the 2-parameter SAR (Test #4) in association with the GR4J and HBV9 models on the 20 snow-affected Alpine catchments. The red crosses indicate mean values.

**Referee's comment**

*10. Table 4: There is a strong drop in the NSE criterion for temperature when going from monthly to daily time steps for IDW and ORK. How this drop can be explained?*

**Authors' response and modifications to manuscript**

The drop in the NSE criterion for temperature was in fact when going from monthly (or daily) to yearly time scale for IDW and ORK. As it can be seen from the following figure 5, the NSE criterion is very sensitive to the number of considered time step, and further on the range of sampled temperatures (which is quite different at the yearly versus monthly time step). As a result, the NSE values between the different methods should be compared only for a given time scale, and not in between time scales.

On the opposite, the RMSE criterion (which was used as objective function in the JK cross-validation) is better representative for the comparison of temperature (whatever the time scale) since units are directly comparable. Following the referee comment, and as NSE values seemed to cause confusion in the result interpretation, only the RMSE values are now presented in Table 4.

[Figure]

**Fig. 5** Cross-validation of the IDW and IED methods against (a) yearly and (b) monthly series from temperature gauges over the period 2000–2016. The NSE criterion is very sensitive to the number of considered time step, and further on the range of sampled temperatures (which is quite different at the yearly versus monthly time step). As a result, there is drop in NSE values for temperature when going from monthly to yearly time step, particularly with the IDW method. Therefore, the NSE values between the different methods should be compared only for a given time step, and not in between time steps. On the opposite, the RMSE criterion (used as objective function for cross-validation) is better representative for the comparison of temperature whatever the time step.

**Referee's comment**

*11. L472-476: I think this result is the consequence of using WB in the objective function. As mentioned above, this constraint is artificial and potentially counterproductive for the efficiency of the model.*

**Authors' response and modifications to manuscript**

This result was indeed the consequence of using WB in the objective function. Please note however that this constraint was not counterproductive for the efficiency of the model as it can be seen clearly from Figures 1 and 2 of the revision notes: with or without WB in the objective function, the hydrological predictions are significantly improved as regards to the efficiency criterions when using a SAR targeting for the temperature and precipitation lapse rates (Tests #4, #5 and #6).

Following the referee comment, a more classical objective function was used (i.e. without the WB term in the OF). Obviously, the water balance was not closed systematically and it was not interesting anymore to present the Budyko graphs for the different tests. The Budyko graphs and associated comments were therefore removed from the manuscript.

**Referee's comment**

*12. L510-516: I find this a bit contradictory with the WB constraint. If the author makes the hypothesis that underground water exchanges between catchments may play a key role, why does the author constrain water balance not to account for such exchanges in the optimization phase?*

**Authors' response and modifications to manuscript**

I do not understand why this comment would be contradictory with the WB constraint. As explained above, inter-catchment groundwater exchanges (IGE) are not the only reason why the water balance may not be closed in the Budyko sense. Other reasons (maybe more important) may play a key role such as errors in the precipitation volumes or wrong estimate of potential evapotranspiration. Since the paper deals with the optimization of temperature (impacting snow accumulation and melt, but also evapotranspiration estimates) and precipitation gradients, constraining the water balance in the objective function aimed mainly at enhancing the parameter identifiability (see Fig. 3 of the revision note) without deteriorating the modelling efficiency (see Figs. 1 & 2 of the revision note). While the HBV model considers the catchments as closed systems, GR4J allows potential IGE via its X2 parameter. Therefore, the Budyko constraint left room for not completely close the water balance with GR4J, as it was commented in the submitted paper.

  However, since sensitivity analyses to the objective function are far beyond the paper issue and because other readers may not be convinced by the proposed constraint without an in-deep demonstration, I renounced to the WB constraint in the objective function (see answers to the referee comment #9) and I re-run all the simulations with a more classical *OF*. Figures and comments were changed accordingly. Please note that it did not change the main findings of the paper.

  The following paragraph (and associated new table) was also added in the section 5.3 (Identifiability of the parameters) to further discuss on the IGE issue and suggest the findings of the initial submission using the WB term in the OF:

"…Equifinality is also reduced in Tests #4–6 for the parameters controlling runoff generation and routing (*X1*, *X3* and *X4*). On the opposite, the parameter of the inter-catchment groundwater flows (*X2*) is poorly identifiable with variation coefficients of 24.8%, 20.3% and 143.1% with Test #4, Test #5 and Test #6, respectively. This suggests that inter-catchment groundwater exchanges (IGE) do not play a key role in the studied catchments. Indeed, fixing *X2* to a value of 0 (i.e. without potential IGE) with an alternative GR3J model provided similar mean validation efficiency on the set of catchments as compared to the GR4J associated with the 2-parameter SAR (Table 7). However, other objective functions may result in other findings as far as IGE are concerned. For instance, additional tests (not shown here for brevity sake) confirmed that it was possible to greatly reduce the X2 equifinality without decreasing the model efficiency by adding a water balance term in the objective function to constrain the proportion of years respecting the water and energy balance in the Turc-Budyko non-dimensional graph (see Andréassian and Perrin, 2012). These tests suggested that it remains important to explicitly represent inter-catchment groundwater transfers in association with correcting or scaling factors applied to the precipitation input data to render the distribution between evapotranspiration, streamflow and underground fluxes more realistic, as already reported by Le Moine et al. (2007)."

**Table** Mean validation efficiency on the set of 20 catchments with the GR4J model and the GR3J model in association with the 2-parameter SAR.

| Model | Total number of free parameters | Mean $NSE_{SNOW}$ | Mean $NSE_Q$ | Mean $NSE_{lnQ}$ | Mean $VE_C$ |
|---|---|---|---|---|---|
| 2-parameter SAR/GR4J | 6 (2 + 4) | 0.86 | 0.79 | 0.82 | 0.95 |
| 2-parameter SAR/GR3J | 5 (2 + 3) | 0.86 | 0.78 | 0.81 | 0.94 |

**Referee's comment**

*13. Fig. 8 is interesting. However there are some cases which reveal that the optimum is probably outside the preset parameter range. This is typically the case for Test#1 for parameters X1 to X3. Therefore the ranges should be extended.*

**Authors' response and modifications to manuscript**

I think the referee comment is too categorical here. For Test#1 (and only for Test#1), the parameters *X1* to *X3* indeed reached the maximum allowed range. Please note however that Test#1 serves as a benchmark. As explained in Table 5 and in the text, it differs from the other tests because no elevation dependency in the *T* and *P* inputs are considered. As a result, hydrologic predictions with Test#1 are significantly (and rather logically) outperformed by the other approaches accounting for elevation-dependency (see e.g. Figures 1 and 2 in the revision note). Extending the range of the parameters would be both poorly efficient in improving the simulations and incorrect from a numerical point of view. The referee has to be aware that the parameter ranges were preset to values recommended by the models' authors (Perrin et al., 2003 for GR4J and Beck et al., 2016 HBV9). They have been found after numerous simulations in very different contexts and can be judged as large enough. By the way, it can be seen in Figure 3 of the revision note, that no parameter limits are reached in the other tests, thus suggesting that the preset parameter range are adequate.

To address the referee comment, I only extended the range of the *X1* parameter (from 10-1000 mm to 0-1500 mm) of GR4J to ensure a better correspondence with the UZL parameter range of HBV9. All simulations were re-run with this new range (and also with an objective function without WB, see answer to the referee's comment #9), and Figures and comments were modified accordingly. Please note that this did not change the results (see Figure 1 of the revision note) and the parameters X2 and X3 still reached the maximum allowed range (see Figure 3 in the revision note) with Test#1 (and only for Test #1) for the reasons explained above. The following comment was also added in the beginning of section 5.3:

"…The maximum allowed parameter range is only reached for the parameters *X1* and *X2* with Test #1. This test differs from the others because no elevation dependency in the *T* and *P* inputs are considered. Consequently, hydrologic predictions of Test #1 are significantly outperformed by the other approaches. Extending the parameter ranges beyond the tested values would be both poorly efficient in improving the simulations and incorrect from a numerical point of view since they were set to values recommended by the models' authors. Moreover, no parameter limits were reached in the other tests, thus suggesting that the parameter ranges are adequate…"

**Referee's comment**

*Cited references:*

*Besic, N., et al. (2014). "Calibration of a distributed SWE model using MODIS snow cover maps and in situ measurements." Remote Sensing Letters 5(3): 230-239.*

*Henn, B., et al. (2016). "Combining snow, streamflow, and precipitation gauge observations to infer basin-mean precipitation." Water Resour. Res. 52(11): 8700-8723.*

*Le Moine, N., et al. (2015). "Hydrologically Aided Interpolation of Daily Precipitation and Temperature Fields in a Mesoscale Alpine Catchment." J. Hydrometeorol. 16(6):2595-2618.*

*Leleu, I., et al. (2014). "Re-founding the national information system designed to manage and give access to hydrometric data." La Houille Blanche(1): 25-32.*

*Naseer, A., et al. (2019). "Distributed Hydrological Modeling Framework for Quantitative and Spatial Bias Correction for Rainfall, Snowfall, and Mixed-Phase Precipitation Using Vertical Profile of Temperature." J. Geophys. Res.-Atmos. 124(9): 4985-5009.*

*Rahman, K., et al. (2014). "Streamflow response to regional climate model output in the mountainous watershed: a case study from the Swiss Alps." Environmental Earth Sciences 72(11): 4357-4369.*

*Riboust, P., et al. (2019). "Revisiting a simple degree-day model for integrating satellite data: implementation of SWE-SCA hystereses." Journal of Hydrology and Hydromechanics 67(1): 70-81.*

*Zhang, F., et al. (2013). "Snow cover and runoff modelling in a high mountain catchment with scarce data: effects of temperature and precipitation parameters." Hydrol. Processes 29(1): 52-65.*

**Authors' response and modifications to manuscript**

Most of the proposed references were judged useful. Therefore, they were cited in the text and added to the reference list, expect that of Leleu et al. (2014) which I could not find.